# Intraspecific host variation plays a key role in virus community assembly

Suvi Sallinen [1,3✉], Anna Norberg [2,3], Hanna Susi [1] & Anna-Liisa Laine [1,2]

Infection by multiple pathogens of the same host is ubiquitous in both natural and managed habitats. While intraspecific variation in disease resistance is known to affect pathogen occurrence, how differences among host genotypes affect the assembly of pathogen communities remains untested. In our experiment using cloned replicates of naive *Plantago lanceolata* plants as sentinels during a seasonal virus epidemic, we find non-random co-occurrence patterns of five focal viruses. Using joint species distribution modelling, we attribute the non-random virus occurrence patterns primarily to differences among host genotypes and local population context. Our results show that intraspecific variation among host genotypes may play a large, previously unquantified role in pathogen community structure.

[1] Organismal and Evolutionary Biology Research Programme, Viikinkaari 1 (PO box 65), FI-00014, University of Helsinki, Helsinki, Finland. [2] Department of Evolutionary Biology and Environmental Studies, University of Zürich, CH-8067 Zürich, Switzerland. [3]These authors contributed equally: Suvi Sallinen, Anna Norberg. ✉email: suvi.sallinen@helsinki.fi

Parasites constitute the majority of biological diversity on our planet[1–4], and they influence both the demography and evolution of their host populations[5–7]. Host susceptibility, pathogen infectivity, and environmental favourability have been identified as the corner stones of disease within the disease triangle framework[8]. However, it is becoming increasingly clear that multiple infections within individuals are abundant[3], and have the potential to change the evolutionary and epidemiological trajectories of pathogens[9]. Consequently, accounting for the diversity of infection is necessary to understand and predict disease dynamics and costs of infection for the host.

Understanding the determinants of the assembly and composition of pathogen communities is one of the key challenges in disease biology today. As a challenge it is analogous to the long-standing debate on the relative importance of biotic interactions versus external drivers of community dynamics. While some theories suggest species interactions to structure biological communities[10,11], others highlight the importance of environmental drivers, including stress and disturbance on community dynamics[12,13]. To date, disentangling biotic processes from the abiotic ones has remained challenging[14]. In recent years, pathogens are increasingly studied within a community ecological framework[15–19]. Environmental variables and wider landscape context, such as human management, are linked to infection load, parasite diversity, and coinfection prevalence across multiple spatial scales[18,20–26]. The composition of parasite communities has also been linked to pathogen transmission mode, degree of host specialty, and life-cycle complexity[27–29], as well as host history, phylogeny, geographical range, longevity, and growth strategy[30–38]. High parasite prevalence itself is a strong predictor of coinfections[9,39]. For vector-borne diseases, positive co-occurrence is common for pathogens that share a vector or transmission site, or when vectors show preference for already infected individuals[15,40,41].

Co-occurrence of pathogens among host individuals is often non-random and coinfections can reach unexpectedly high levels[15,20,23,42–44]. One of the key challenges is to determine how biotic interactions between hosts and their pathogens themselves shape these distributions. Under the community ecological framework, a host can be viewed as a resource patch and its resistance as a local filter that determines the pathogen community within that host[18]. Hosts are resistant against most pathogen species they encounter[45], and even for pathogens capable of infecting a host species, there is often considerable variation among individuals in their susceptibility[7,46–50]. The effect of intraspecific variation in disease resistance on the dynamics of individual pathogens is well described[51–54]. However, the importance of intraspecific host resistance variation for community assembly and diversity of species that exploit the host is only beginning to gain attention[55]. Due to allocation costs associated with genetically-based resistance, a host resistant against a particular pathogen may be susceptible to others[56,57]. On the other hand, limited evidence suggests that the same resistance loci may provide protection against several different pathogens[58]. Pathogens attacking the same host may also compete for host resources (resource-mediated interaction), or interact via elicited host immune responses[59]. Induced immunity by a first arriving pathogen may change the resistance phenotype, as immuno-suppression of the host by the first arriving pathogen may facilitate establishment and replication of later arriving pathogens[60–63]. On the other hand, cross-reactive immune responses elicited by the first parasite have the potential to suppress the success of later arriving parasites[63,64]. These biotic interactions could result in non-random pathogen co-occurrence patterns across host genotypes. Variation in host resistance may be spatially structured with pronounced differences in resistance observed among host populations[53] and regions[65]. Such spatially structured resistance variation may also drive spatially structured co-occurrence patterns of pathogens exploiting the same host. Whether the host genotype is indeed a strong determinant of within-host parasite communities in the wild, and what the consequences of these within-host parasite community assembly processes are for host populations, remain unanswered[18,66].

Here, we study the importance of the host genotype in determining the structure of within-host virus communities. Viruses are in principle obligate parasites as they require a host for reproduction. A growing body of evidence has demonstrated that consequences of virus infection can shift along the pathogenic–mutualistic-continuum, even for the same interaction[67,68], and visually asymptomatic infections are common in wild plants[3]. Using cloned replicates of naive *Plantago lanceolata* plants as sentinel traps placed in natural populations during a seasonal epidemic of viruses, we can tease apart the role of the host genotype from drivers that affect the distribution of viruses within the local population context, which may include environmental variation, the local disease pool, host population structure and history, as well as local vector communities. Moreover, we aim to understand how biotic interactions among the viruses[59–64] influence their community assembly.

We characterize the establishing virus communities using PCR detection[69]. We first test whether the viruses occur in the same sentinel plant more often than would be expected based on their frequencies alone. In other words, we test whether virus co-occurrence patterns differ from expectations of a random distribution. We then employ a joint species distribution modelling (JSDM) framework[70], that allows us to tease apart the effect of local population context (consisting of unmeasured environmental variation as well as host population structure and history) on virus (co-)occurrences from host plant characteristics and host genotype. We can account for the shared environmental responses of the target species, which makes the model a robust method also for sparse data[71]. Using this approach, we are also able to capture signals of possible biotic community assembly processes from virus-to-virus association matrices after controlling for shared environmental responses of the viruses. The performance of JSDMs in relation to traditional, single-species distribution modelling (SDM) methods has recently been validated[72]. The application of these kinds of multivariate statistical tools— typically used in community ecological analysis—to parasite data has the potential to reveal new insights of the determinants of parasite community assembly and composition[73,74].

In this study, we ask: (1) Do we see more (or less, respectively) co-occurrences between the viruses than what would be expected solely based on their frequencies?; (2) Does the local population context affect the virus community composition?; (3) Do host genotypes differ in the virus communities they acquire, suggesting genotype-level variation in overall sensitivity to infection?; (4) *After* accounting for the aforementioned effects (2–3) of the local population context and plant host characteristics (including the host genotype), is there evidence of *residual* virus co-occurrence patterns across the entire data indicative of competitive or facilitative virus interactions?; and (5) Do these residual co-occurrence patterns vary among host genotypes indicating genotype-specific resistance responses affecting virus community structure? Our results indicate that while the population context also drives virus community assembly, host genotypes vary in the virus communities they acquire.

## Results

**Detection of viruses in the field experiment.** Out of the 320 sentinel host plants, 68% were hosts to at least one virus over the study period. Three viruses were clearly more common

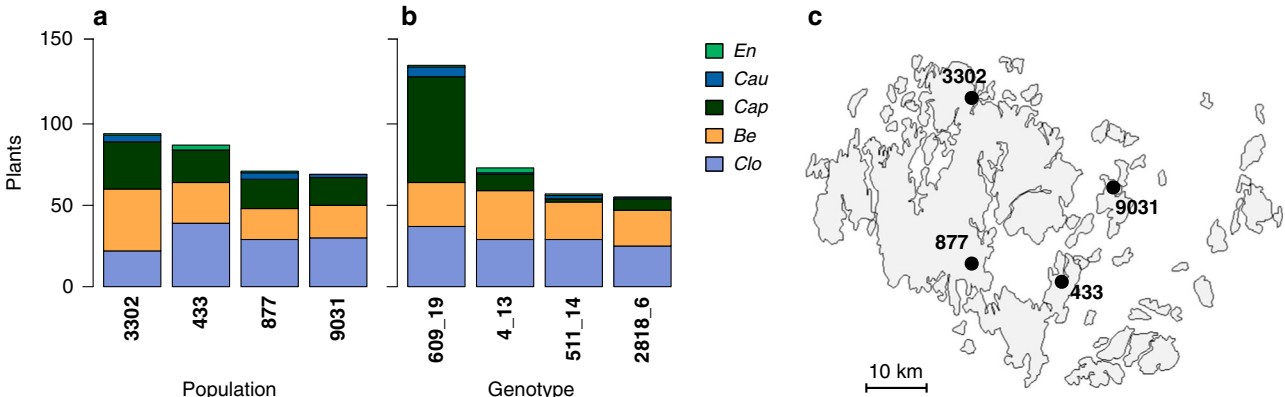

**Fig. 1 Virus infections in sentinel plants.** Infections are plotted by population (**a**) and genotype (**b**), and the locations of the study populations in the field experiment in the Åland Islands (**c**). The genotypes and populations are ordered from left to right according to decreasing overall number of infections. '*Clo*' refers to *Plantago closterovirus*, '*Be*' to *Plantago betapartitivirus*, '*Cap*' to *Plantago lanceolate latent virus*, '*Cau*' to *Plantago latent caulimovirus*, and '*En*' refers to *Plantago enamovirus*.

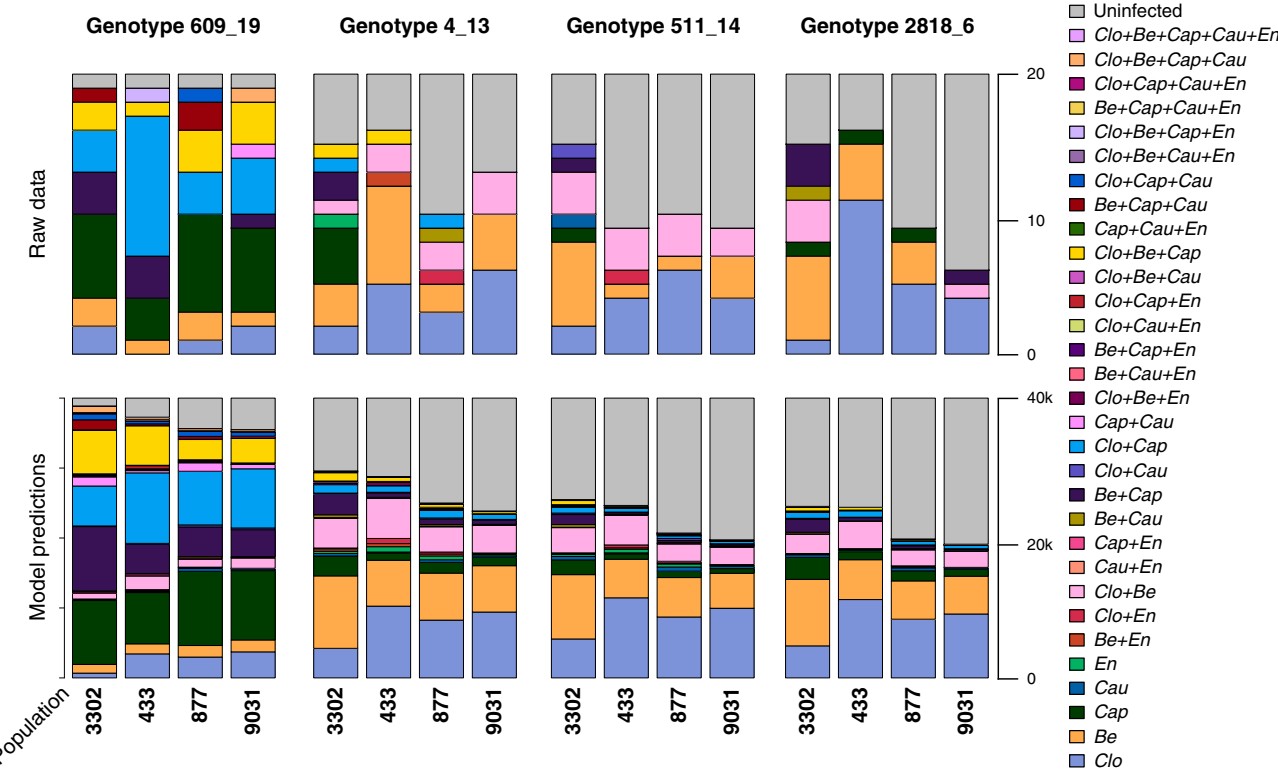

**Fig. 2 (Co-)infections in the original data (upper panel) and predicted coinfections (lower panel) based on the model variant 2, ordered by host genotypes and population, as indicated by the X-axis.** Both the genotypes and populations are ordered with respect to frequency, so that the bars on the left-hand side show the population and genotype with the highest total amount of virus infection. '*Clo*' refers to *Plantago closterovirus*, '*Be*' to *Plantago betapartitivirus*, '*Cap*' to *Plantago lanceolate latent virus*, '*Cau*' to *Plantago latent caulimovirus*, and '*En*' refers to *Plantago enamovirus*. The total number of plants in the upper panel is 20, whereas in the lower panel the total number is simulated plants is 20 (original number of plants) times 2000 (number of MCMC iterations used for the simulation), resulting in 40,000 simulated plants.

in the sentinel plants: closterovirus in 120 individuals, betapartitivirus in 102 individuals, and capulavirus in 84 individuals; while caulimovirus and enamovirus were rare: in 10 and 5 individuals, respectively (Figs. 1a and 2). Out of the 217 infected individuals, 49 (23%) hosted more than one virus, and in total, we found 17 virus combinations, ranging from single infections to four of the five viruses found in the same plant (Fig. 2). Both overall virus prevalence and the composition of virus communities varied among plant genotypes and plant populations (Figs. 1a, b and 2).

**Analysis of virus co-occurrence.** We found significant non-random positive co-occurrences between species pairs capulavirus and caulimovirus as well as betapartitivirus and caulimovirus, when we analysed the complete data set (Fig. 3). When we analysed the co-occurrences separately for each host plant genotype, we found positive co-occurrences between betapartitivirus and caulimovirus on genotype 609_19, as well as between betapartitivirus and capulavirus on genotype 2818_6. We also found negative co-occurrences between betapartitivirus and closterovirus, as well as capulavirus and closterovirus on plant genotype

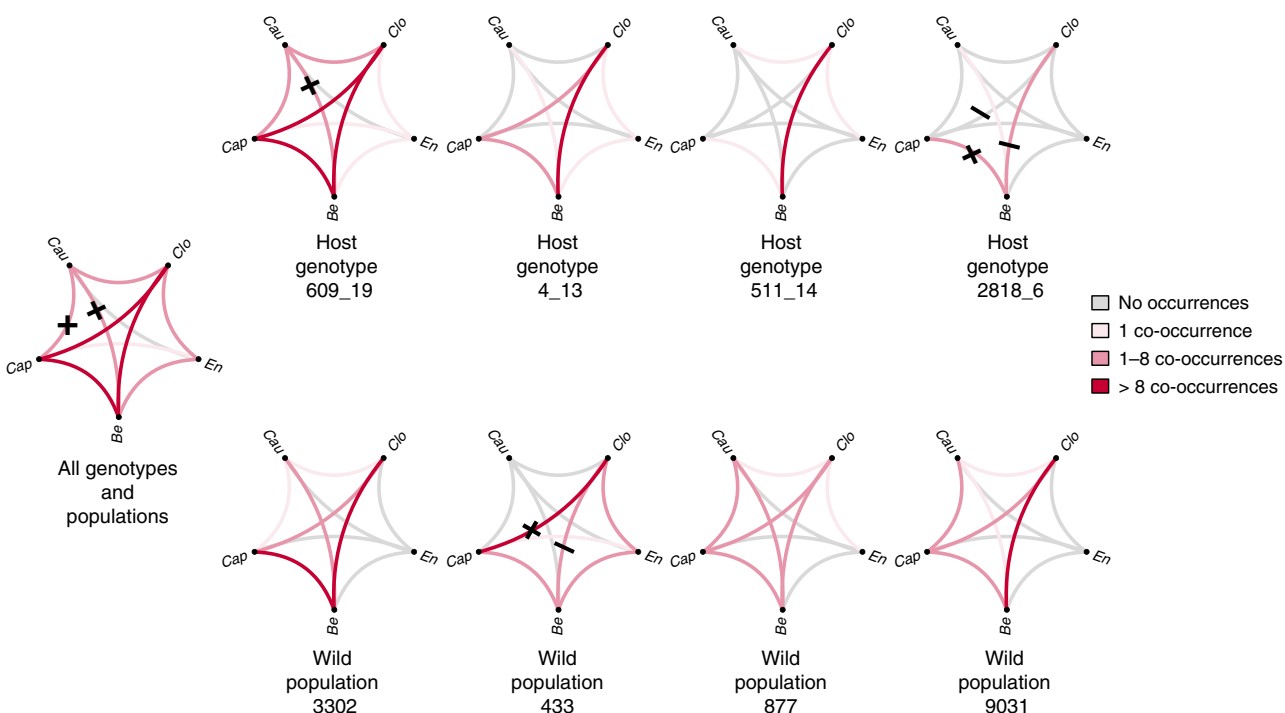

**Fig. 3 Co-occurrences between virus species.** Co-occurrences are shown either in the whole data set (left, with total number of sentinel plants 320), or per plant genotype (upper panels, 80 plants per genotype), or by population (lower panels, 80 plants per population) as denoted by the horizontal axis. The genotypes and populations are ordered from left to right according to decreasing the overall frequency of disease. The plus (and minus) signs denote the pairs, for which the observed values were higher (or lower, respectively) than what would be expected based on their overall frequencies, and for which the probability of this difference was <0.1. The line colours denote the true numbers of co-occurrences between the species, as shown in the legend. 'Clo' refers to *Plantago closterovirus*, 'Be' to *Plantago betapartitivirus*, 'Cap' to *Plantago lanceolate latent virus*, 'Cau' to *Plantago latent caulimovirus*, and 'En' refers to *Plantago enamovirus*. The exact probabilities for the focal pairs are provided in the Supplementary Table 3.

**Table 1 Joint species distribution model variants and their explanatory performance and predictive performance (based on cross-validation), measured by the Tjur $R^2$ coefficient of determination[110] (see 'Methods' and Supplementary information).**

| Model variant | Fixed explanatory variables | Random effects (latent variables) | Explanatory performance | Predictive performance | WAIC |
|---|---|---|---|---|---|
| 1 | Local population context, host size, signs of herbivory | Host plant individual | 0.072 | 0.041 | 2.00 |
| 2 | Host genotype, local population context, host size, signs of herbivory | Host plant individual | 0.16 | 0.11 | 1.78 |
| 3 | Host genotype, local population context, host size, signs of herbivory | Genotype-dependent host plant individual | 0.16 | 0.11 | 1.81 |

2818_6 (Fig. 3). When analysing co-occurrence patterns within each population, we found a significant positive association between capulavirus and closterovirus, and negative association between closterovirus and betapartitivirus in plant population 433. The expected and observed numbers of co-occurrence, as well as the exact probabilities for a greater or smaller number co-occurrences than expected for these species pairs, are provided in Supplementary Table 3.

**Joint species distribution models of virus communities.** The model variants 2 and 3 performed almost equally well, as seen from their performance (Table 1). Model variant 1, excluding host plant genotype as a covariate, was clearly inferior. Model variant 2 also resulted in the smallest WAIC value, implying best predictive power. We did not detect any significant residual co-occurrence patterns between viruses after accounting for the effect of the local population context and host-related variables.

We looked into this with sentinel plant level latent variables that are uniform across sentinel plant genotypes (model variant 2) as well as sentinel plant level latent variables that covary with sentinel plant genotype (model variant 3), and neither of these model variants captured virus co-occurrences with strong statistical support and their explanatory performances did not differ. Based on these results, we decided to consider the simpler model variant 2 as our best model.

The variance partitioning conducted for the model variant 2 revealed sentinel plant genotype to be the most important determinant for virus community composition (42% of variance explained, averaged over species; Fig. 4), followed by the local population context (29%; Fig. 4). The importance of variables differed between the viruses. Plant genotype explained most of the variation for capula- and caulimoviruses, while for enamovirus the sentinel plant genotype and the population context were almost equally important. For clostero- and

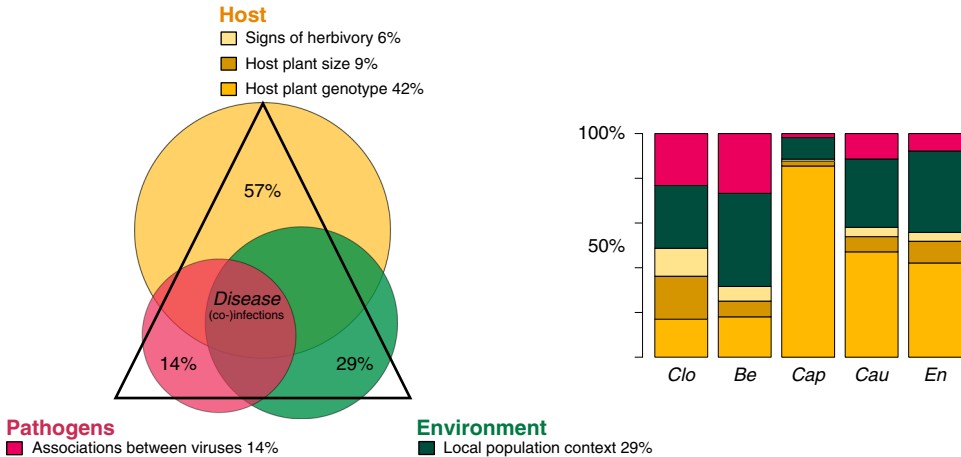

**Fig. 4 Partitioning of the variance explained by model variant 2 (Table 1).** The diagram overlays the average proportions (over species) of variance explained by different groups of explanatory variables (out of the total variation explained by the model) and the concept of the disease triangle. The legend labels denote the different variables for which the partitioning is calculated, and the percentages indicate the mean values for the whole community. The barplot gives these results separately for each virus: the horizontal axis shows the focal five viruses (ordered from left to right according to their decreasing overall infection rate) and the vertical axis shows the proportion of variance explained. 'Clo' refers to *Plantago closterovirus*, 'Be' to *Plantago betapartitivirus*, 'Cap' to *Plantago lanceolate latent virus*, 'Cau' to *Plantago latent caulimovirus*, and 'En' refers to *Plantago enamovirus*.

| Table 2 Regression coefficients for model variant 2 for each virus species. | | | | | |
| --- | --- | --- | --- | --- | --- |
| | **Clo** | **Be** | **Cap** | **Cau** | **En** |
| (Intercept) | **−1.1** | **−1.2** | **−1.8** | **−1.9** | **−2.0** |
| Host plant size | **0.00094** | 0.00042 | 0.00056 | 0.00017 | −0.00038 |
| Signs of herbivory | **0.48** | 0.30 | −0.22 | −0.29 | −0.21 |
| Population 9031 | **0.30** | 0.13 | 0.053 | **−0.47** | 0.27 |
| Population 3302 | −0.20 | **0.57** | **0.62** | 0.18 | −0.084 |
| Population 433 | −0.0070 | −0.091 | −0.14 | −0.26 | −0.58 |
| Genotype 609_19 | 0.12 | 0.12 | **2.47** | **0.67** | 0.067 |
| Genotype 4_13 | 0.017 | 0.23 | **0.46** | −0.011 | 0.35 |
| Genotype 2818_6 | −0.16 | −0.081 | 0.23 | −0.25 | −0.57 |

Posterior mean estimates with statistical support based on the 90% central credible interval are denoted by bold font. 'Clo' refers to *Plantago closterovirus*, 'Be' to *Plantago betapartitivirus*, 'Cap' to *Plantago lanceolate latent virus*, 'Cau' to *Plantago latent caulimovirus*, and 'En' refers to *Plantago enamovirus*.

betapartitiviruses the population context explained more variation in the data than plant genotype.

The importance of the random effect at the level of sentinel plant individuals differed between the viruses, but followed roughly the same pattern: For capula-, caulimo- and enamovirus, the sentinel plant individual random effect was minor, but for clostero and betapartitivirus, its effect was slightly more pronounced (resulting in a total average effect of 14%). However, further inspection revealed that none of the residual correlations between virus species gained strong statistical support. Hence, we see no signal of potential biotic interactions between viruses after taking into account the effects of fixed explanatory variables, i.e. the sentinel plant genotype, size, signs of herbivory and local population context.

As expected, the predicted coinfections based on model variant 2 show similar patterns to what we can see in the raw data (Fig. 2). When examining both the coinfection profiles (Fig. 2), and the posterior mean estimates for the regression coefficients (Table 2), we see that capula- and caulimovirus are much more likely to occur on sentinel plant genotype 609_19 (with posterior mean estimate 2.47 for capula- and 0.67 for caulimovirus, Cap and Cau in Table 2, respectively, that gained strong statistical support based on the 90% central credible interval). Other sentinel plant genotypes were more dominated by single infections of closterovirus and betapartitivirus as well as their co-occurrences. Thus, the overall structure of the virus

communities among plant genotypes was similar regarding the two most prevalent species closterovirus and betapartitivirus, but sentinel plant genotype 609_19 hosted significantly more capulavirus, which consequently also increases the probability of coinfections between capulavirus and other viruses. Regarding caulimovirus, six out of the total ten of its occurrences were together with capulavirus, and all of these co-occurrences were on sentinel plant genotype 609_19. Closterovirus, betapartitivirus and capulavirus are tenfold more prevalent in our data in comparison to caulimovirus and enamovirus, which can be seen in their dominance of the co-occurrence patterns in the community.

Sentinel plant size had a more minor effect on the community structure, as did signs of herbivory (Fig. 4), although both sentinelt plant size and herbivory did have a minor positive effect with strong statistical support on the probability of occurrence of closterovirus (Table 2).

Our result for the same set of model variants fitted with less conservative priors for the latent part of the model show corresponding results to our main variants: model variant 1 is clearly inferior, whereas there is no big difference between variants 2 and 3. With model variants 2 and 3, we are able to detect one association with strong statistical support, between betapartitivirus and caulimovirus. For more details, see our Supplementary information on the joint species distribution modelling.

## Discussion

Understanding how pathogen communities are formed is a key challenge in understanding disease dynamics, as multiple infections can be significant drivers of epidemics as well as pathogen virulence and evolution[9,18,19,75]. The host is expected to be a strong determinant in the formation of pathogen communities, as both theory and controlled experiments have demonstrated host resistance to be a key determinant of disease dynamics[76–80]. Indeed, diversity of resistance in host populations could partly explain non-random co-occurrence patterns of pathogens detected in wild plants[15,20,23,44,46]. In our field experiment using sentinel plants of four genotypes, we found that most of the model-explained variation in virus occurrences was explained by the local population context and sentinel genotype (Fig. 4). Some viruses occurred significantly more or less together than would be expected based on their frequencies in both the full data set as well as when sentinel plant genotypes and local population context were analysed separately.

However, the results of our JSDM modelling (Table 1) indicate that the patterns evident in the co-occurrence analysis (Fig. 3) are influenced more clearly by the local population context and host genotype variation than by direct or indirect biotic interactions among the viruses. While disentangling host genotypic effects from other factors affecting pathogen communities has remained challenging, we were able to uncover the roles of these determinants of virus communities in wild hosts using naive sentinel plants in wild plant populations.

Of the total amount of variation explained with our best model variant, the population context explained within-host virus communities to a large extent, although the proportion of explained variation varied among the viruses (Fig. 4). Drivers that could vary among our plant population include abiotic variables which we did not explicitly record as many more plant populations would be needed to tease apart relevant variation in local population context for virus communities. These drivers are often found to filter parasites according to their niche preferences from the regional disease pool into local populations, thereby playing a major role in how within-population and -host-parasite communities are formed[20,22,26]. In addition to abiotic variables, the local P. lanceolata populations are likely to differ in biotic factors including plant species community composition and abundance of suitable vectors which may be linked to virus prevalence and diversity[15,20]. The local population context further includes any differences in population dynamics and trajectories, such as historical pathogen pressure, which may vary among these populations[81]. Albeit non-significant, the effect of sentinel plant individual on the (co-)occurrences of the viruses can be attributed to some unmeasured abiotic or individual-related variables, which may influence the (co-)occurrences of the viruses.

While there are multiple studies investigating within-host parasite communities[44,69,73,74,82], to our knowledge the effect of host genotype on the assembly has rarely been tested experimentally in wild systems, or with multiple parasites simultaneously. In our data, sentinel plant genotype accounted for most of the variation in virus occurrences of the total variation explained in the JSDM model. Indeed, both virus occurrence, and the acquired virus communities varied among the four P. lanceolata genotypes. In particular, sentinel plant genotype 609_19 had greater infection prevalence and diversity of viruses than the other genotypes (Fig. 2). As our model controlled for the effect of sentinel plant size and level of herbivore damage, such host genotype-level differences may reflect variation in constitutive resistance, such as resistance genes, among the plant genotypes. The natural P. lanceolata populations in the Åland Islands contain considerable phenotypic variation in resistance against powdery mildew P. plantaginis[83,84], and while resistance against

viruses in this system is not well understood, an exceptionally diverse repertoire of candidate loci (Nucleotide-binding leucine-rich repeat; NLRs) that confer resistance against a broad range of pathogens, have been characterized in P. lanceolata (Laine, personal communication). Uncovering both phenotypic and molecular level virus resistance in this system is an important avenue of future research. Spatially structured variation in resistance is characteristic of natural host-parasite systems[53,85–87], and based on our findings, intraspecific variation in disease resistance in a host population may play a large, previously unquantified role, in the non-random distribution of co-occurring pathogens that have been detected in the previous studies[15,20,23,44,46].

Intraspecific variation in traits other than resistance could also generate the differences we observe. To confirm which traits are involved, future studies should explore in more detail the ecological outcomes of these interactions, and their molecular underpinning. It is highly plausible that the host genotype could indirectly affect virus occurrences via their attractiveness or resistance against vector herbivores[88,89]. Vector preference for infected hosts[41,90] could also influence virus co-occurrence patterns. Transmission mode is often found to be critical for how pathogen communities are formed[15,40,91,92], and reciprocally, the amount of genotypic variation within a host population may explain the abundance and composition of herbivory community present[89].

A community of pathogens could be shaped by both direct and indirect pathogen–pathogen associations: reaction triggered by an earlier arrival could either induce or suppress resistance against later arriving pathogens, or within-host competition could favour one pathogen over the other. Evidence for both negative and positive pathogen–pathogen interactions have been reported in studies of multiple infections[19,59,62,63,93,94]. Although we find both positive and negative co-occurrence patterns among the viruses, these are largely explained by local population context and host genotype. After controlling for these in our model, we do not find strong statistical support for signals of associations among the viruses, as would be expected if arrival by one would decrease or increase the arrival probability of another. Hence, our results do not support the hypothesis that virus–virus interactions —either direct or those mediated by host immunity—would be the key drivers of virus community assembly at the within-host level in this system. However, our sample size could be insufficient to detect such interactions as some of our viruses are rare, and their arrival probability to the sentinel plants is also subject to random processes. In addition, we only accounted for a subset of all possible pathogens infecting plants in this system, thereby potentially missing some influential members of the community. Furthermore, the effects of induced immunity triggered by a first arriving pathogen may be short-lasting[63,95] and, therefore, undetectable with the timescale of this experiment. Induced immunity could play a more important role among viruses of the same genus or strains of the same virus species, where the famous phenomenon of cross-protection is more often recorded[63,96] and as is predicted by theory[75]. Given that the variants with less conservative priors detected a significant positive interaction between betapartitivirus and caulimovirus, we conclude that our study design was successful in capturing the effects of the host genotype, but larger-scale investigations would be required to detect signals of virus–virus interactions.

In our experimental design we kept the plants in their pots which meant these plants experienced different rooting environments than the wild plants but allowed us to standardize some factors (e.g., soil medium). However, this approached allowed us to control for this level of variation in our data. Our approach may have affected vector preferences as visual presentation of the plants, in addition to other cues, is important for vector

dynamics[90]. Nonetheless, transmission of all five focal viruses to the sentinel plants did occur. Whether the virus prevalences we detected with our approach are in line with infections of wild plants is difficult to assess, given that virus prevalences vary greatly among populations in the Åland Islands (0–64%)[69]. Overall, our study does not only highlight the importance of the host genotype, but also the need for further research on other aspects of virus ecology. Although we have placed the current work into a context of pathogens, viruses may also have neutral or positive effects on the hosts despite their parasitic lifestyle[67]. While knowledge of virus diversity and roles of viruses in wild populations is increasing[3,44,67], research at the community scale remains scarce[18].

Here, we have quantified the importance of intraspecific host plant variation on how within-host virus communities assemble by using sentinel plants in natural populations during a seasonal epidemic, which allows teasing this factor apart from other drivers of virus occurrence. Applying JSDMs to interpret the effects of host genotype and local population context, we find that while the population context has a strong influence on virus communities within individual hosts, not accounting for the host genotype might underestimate the role host genotypes have in generating variation in pathogen communities. Such variation in within-host pathogen diversity may have far reaching implications for all key aspects of disease: transmission, virulence suffered by the host, and pathogen evolution. With these results, we are one step closer to binding together the different spatial scales and processes that underpin pathogen metacommunities.

## Methods

**Study species.** *Plantago lanceolata* is a globally occurring perennial herbaceous plant[97]. It is an obligate outcrosser with wind-dispersed pollen, also capable of vegetative reproduction[97]. In the Åland Islands, SW of Finland, it typically grows on dry meadows, forming a network of approximately 4000 small connected populations[81]. The size and location of the populations have been monitored since the early 1990s as a part of the metapopulation studies of the Glanville fritillary butterfly and powdery mildew *Podosphaera plantaginis*[81,98]. In the Åland Islands, *P. lanceolata* also hosts a diverse community of viruses that vary in their occurrence among *P. lanceolata* populations and among the individuals within populations[69]. We used five recently characterized viruses from the Åland Islands, to study within-host viral communities[69]: *Plantago lanceolata latent virus* in genus Capulavirus and *Plantago lanceolata caulimovirus* in genus Caulimovirus with DNA-genomes, and *Plantago betapartitivirus* in genus Betapartitivirus, *Plantago enamovirus* in genus Enamovirus, and *Plantago closterovirus* in genus Closterovirus with RNA-genomes. The viruses are hereafter referred to by their genus for understandability. These viruses were initially identified from *P. lanceolata* in the Åland Islands by sequencing plant small RNAs[69]. Plants use RNA-silencing mechanism and produce short interfering RNA (SiRNA) molecules in a defense response against viral infection[99]. Hence, these viruses trigger an active defense response in *P. lanceolata*. Also, although not directly demonstrating their pathogenic nature, Susi et al.[69] found that plants with virotic symptoms (necrotic spots/ yellow colour) are more likely to carry a virus infection. Currently, the detailed transmission dynamics and vector species, as well as the viruses' distribution outside the Åland Islands remain unknown. More detailed information of the virus families is compiled in Supplementary Table 1.

**Field experiment with sentinel plants of different genotypes.** To study the effect of plant host genotype on the variation of within-host virus communities, we set up an experiment using sentinel trap plants in natural populations of *P. lanceolata* in the Åland Islands. To obtain genetically uniform plant material, we cloned four greenhouse-grown maternal *P. lanceolata* plants into 80 replicates each. The maternal plants originate from natural *P. lanceolata* populations in the Åland Islands, and were grown from seeds in an insect free greenhouse at the University of Helsinki. The plants are expected to represent four different genotypes (ID:s 609_19, 4_13, 511_14, 2929_6), as their maternal plants originated from distant populations 7–40 kilometres apart. Their resistance against viruses is currently unknown, but they represent different mildew resistance phenotypes as has been confirmed during laboratory maintenance of *P. plantaginis*. The maternal plant individuals used in the experiment were confirmed to be free of target viruses, that would have been the result of seed-borne infection, by PCR-testing using specific primers. Each maternal plant was cloned into 80 replicates by placing maternal plants on pots containing vermiculate and kept on a tray containing fertilized water.

After one month, the roots grown from the maternal plant's pot through to the vermiculate were cut. After another month, new plants shooting from the cut roots in the vermiculate were separated and individually planted into 10 cm × 10 cm pots containing an equal amount of sand and potting soil. After two additional months in the greenhouse, during the last week of May 2017, the plants were taken to the Åland Islands and placed into four *P. lanceolata* populations (ID:s 877, 9031, 433, 3302; Fig. 1c). The populations were selected for the study as they represent different parts of the Åland Islands, were remote to humans, and large enough to host a field-experiment. These populations were different from the ones the maternal plants used for cloning originated from. These four populations were included in the analyses as a categorical variable to capture 'local population context' (local temperature, vectors, plant communities, etc.) that may influence virus distributions among *P. lanceolata* populations in the Åland Islands.

Twenty replicates of each sentinel plant genotype were placed into each of the four *P. lanceolata* populations resulting in 80 plants per population, and 320 plants altogether. The plants were kept in their pots for the duration of the experiment, and they were placed in a random order among natural vegetation and reshuffled three times per week to avoid within-population spatial effects. The plants were kept separated from the local soil on plastic freezer boxes and watered when necessary. Signs of herbivory (holes, bitemarks, and thrip damage) were recorded after two weeks of exposure, and again after seven weeks of exposure. Plant size was measured during the first week of exposure by counting the number of leaves, and by measuring the length and width of the longest leaf. Based on these measurements we calculated plant size by using the equation $n \times A$, where $n$ is the number of leaves, and leaf area $A$ is calculated using the equation of ellipse area: $A = \pi ab$, where $a$ is a half axis of the width of the longest leaf, and $b$ is the half axis of the length of the longest leaf. For those 13 plants missing measurement data, an average over all recorded values for all plants was used, in order to not to lose any virus occurrence data from the analysis.

**Nucleic acid extractions and virus detections with PCR.** To detect the viruses infecting plants during the growing season, leaf samples were collected for nuclear acid extractions after two weeks and again after 7 weeks of exposure to the natural virus and vector communities. Samples were collected from a single leaf of similar age (young but large enough for sampling) from each plant. For DNA extraction, we collected a 1 cm² piece of leaf from each plant. Samples were stored in −20 °C until DNA extraction with E.Z.N.A. Plant Kit (Omega Biotek, USA) at the Institute of Biotechnology at University of Helsinki. For RNA extractions, 3 cm² leaf samples were collected, immediately deep-frozen in liquid nitrogen, and stored in −80 °C before RNA-extraction. Total RNA was extracted using phenol-chloroform extraction with a modified method from Chang et al.[100]. Two additional phenol cleaning steps prior chloroform cleaning of the RNA were performed. In the additional cleaning steps, we used 800 ml of equal volumes of phenol solution (pH 4.5) and chloroform-isoamylalcohol, mixed with isolation buffer containing the sample, vortexed, and centrifuged for phase-separation in 14 800 rpm for 15 min. For the PCR detection of the RNA viruses, RNA was translated into cDNA. For reverse transcription, we used 2 ng of total RNA, mixed with 2 μl random hexamer primers (Promega) and sterile nuclease free water in 17,125 μL volume incubated for 5 min in 70 °C. Subsequently, 1 μL Moloney Murine Leukemia Virus Reverse Transcriptase (M-MLV RT; Promega Corporation, USA), 5 μL M-MLV RT buffer, 1.25 μL of dNTP (10 mM) mix, and 0.625 μL of RiboLock RNaseinhibitor were added and the 37.41 μL reaction mix was incubated in 37 °C for 60 min. For virus detection PCR, we used specific primers[69,101] as well as two additional primer pairs for capulavirus (PiLVi2_forward_1 5′-GTGTTTAACAATGAAGT GAGCC-3′ and PiLVi2_reverse_4 5′-AATCCATCCACACATCCAATC-3′) and caulimovirus (forward primer 5′-AGGAGATGCCCATACTTTACC-3′ and reverse primer 5′-GACTTGCCAGAACCTGATTTAC-3′). PCR reactions to detect viruses were performed in final volume of 10 μL containing of 1–3 μL of DNA or cDNA, and GoTaq Green® polymerase 5x Mastermix (Promega Corporation, USA) according to manufacturer's instructions. Samples were subjected to initial denaturation in 95 °C for 2 min, following 35 cycles of denaturation in 95 °C for 40 s, annealing 53–60 °C for 40 s, and extension 72 °C for 1 min with a final extension step of 72 °C for 5 min. The full protocol with virus specific PCR conditions is described in the Supplement (section 'PCR-detection of viruses'). The amplicons were resolved on a 1.2–1.5% agarose gel and visualized using Gel Doc XR System (Bio-Rad Laboratories, Inc., USA).

**Statistical analysis.** For all the statistical analysis, we pooled the detected occurrences of the five focal viruses over the two timepoints of sampling by collapsing the occurrence data so that each sentinel plant had one observed virus community. Only when a sentinel plant had not been infected by a certain virus in either of the timepoints accounted as an absence of the virus while infection in one or both timepoints was accounted for as virus presence. To understand whether the co-occurrence of viruses differs from expected co-occurrences calculated solely from the prevalences of these viruses, we first analysed the co-occurrence patterns both in the full data set as well as separately for each sentinel host genotype and plant population (Fig. 3). We used the R package 'cooccur'[102] and its identically named function, and applied a probabilistic model[103] which calculates expected frequencies of species co-occurrences based on a distribution of random, independent species. By comparing the expected and observed co-occurrences the

applied algorithm gives the probabilities of co-occurrence greater than or less than what is observed in the data analytically, without relying on randomisations or test statistics, under the condition that the probability of occurrence for a species at each sentinel plant is equal to its observed frequency among all the sentinel plants, i.e. in this case the prevalence of the virus[103].

For addressing our study questions about the effects of host genotype and characteristics as well as local population context on the (co-)occurrence patterns of the viruses, as well as the possible signals of biotic interactions between the viruses on virus community assembly, we applied a joint species distribution modelling (JSDM) framework 'Hierarchical Modelling of Species Communities' (HMSC[104]), which is a multivariate Bayesian hierarchical generalised linear latent variable model. Essentially, HMSC is a multivariate generalised linear model, enabling the modelling of the whole community of viruses as opposed to fitting individual single-species distribution models[105]. In addition, HMSC is a latent variable model[70]. Latent variable models include unobserved, i.e. latent predictors, which are typically included to model correlation, or to account for missing predictors[70]. Hence, in this context, the latent variables are random effects that model the co-occurrences between species due to either biotic interactions or some other effects not included in the fixed part of the model, such as unmeasured effects of the environment. For a more detailed description of JSDMs and latent variable models, please see the comprehensive review by Warton et al.[70].

The structure of the HMSC modelling framework is described in detail by Ovaskainen et al.[104,106], with connections to community ecological theory and case studies. In our study, we modelled the virus community, denoted by the $n \times n_s$ matrix $Y$ of virus occurrences, comprising of individual components $y_{ij}$, denoting virus $j = 1, ..., n_s$, where $n_s = 5$, on host plant $i = 1, ..., n$, where $n = 320$, with probit regression

$$y_{ij} = 1_{L_{ij} + \varepsilon_{ij} > 0}$$

$$L_{ij} = L_{ij}^F + L_{ij}^R$$

where $\varepsilon_{ij} \sim N(0,1)$, $L_{ij}$ is the linear predictor for the occurrence of virus $j$ on sentinel plant $i$, which is further divided to fixed ($L_{ij}^F$) and random ($L_{ij}^R$) parts. The fixed effects F model the influence of the local population context and the influence of the sentinel plant characteristics. The random effect R models the residual variation in virus occurrences at the level of individual sentinel plants, that cannot be attributed to the above-described responses of the viruses to the fixed covariates. For exact formulation how the different components are modelled, with corresponding notation, please see Ovaskainen et al.[106].

Briefly, following the compact matrix notation of Chapter 7.3.2 in Ovaskainen et al.[106], we model the $n \times n_s$ community matrix of viruses $Y$ with a $n \times n_s$ matrix $L$ of all linear predictors $L_{ij}$ for all species and sentinel plants, as $L = L^F + L^R$. The matrix of fixed effects can be further decomposed as $L^F = XB$, where $X$ is the $n \times n_c$ matrix of environmental covariates, and $B$ is the $n_c \times n_s$ matrix of regression coefficients, i.e. species responses to the covariates, and $n_c$ is the total amount of covariates included in the model. Because the environmental covariates $X$ are known and given as input for the model (Table 1), only the species responses $B$ are estimated. Analogously, the matrix of random effects can be decomposed as $L^R = H\Lambda$. Here, $H$ is the $n \times n_f$ matrix of latent factors, or site loadings, and $\Lambda$ is the $n_f \times n_s$ matrix of latent factor loadings, or, where $n_f$ is the number of latent factors. Both the site $H$ and $\Lambda$ are estimated, as is the number of latent factors $n_f$. The species loadings $\Lambda$ can then be translated into residual associations between virus species by transforming them into covariation between species as $\Omega = \Lambda^T\Lambda$, and further into correlations.

We fitted three JSDM variants to the data by varying the way the sentinelt plant genotype was included in the model (Table 1). As explanatory variables (denoted by matrix $X$ in ref. [71]) we used the local plant population context (categorical variable with four classes), which is a proxy for the plant population-level effects, such as variation in abiotic conditions, vector communities, and disease pool (categorical variable with four classes); and at the level of the sentinel host plants, we include the plant size (a continuous variable), signs of herbivory (a categorical variable with two classes; yes/no), as well the genotype of the sentinel host plant (a categorical variable with four classes). To examine the residual co-occurrence patterns among hosts, we also included the sentinel plant individual as a latent variable random effect.

First, we fitted a model with only the local population context, plant size and signs of herbivory (variant 1) as fixed explanatory variables $X$. Then, we fitted a model including also the sentinel plant genotype, i.e. the full set of fixed explanatory variables (variant 2). With both of these model variants (1 and 2), we included random effects at the level of sentinel plants individuals. Finally, we fitted a model with the same full set of fixed explanatory variables $X$ as with model variant 2, but we modified the random effects by allowing these residual patterns to covary with the sentinel plant genotype (variant 3), details of which are explained by Tikhonov et al.[107]. In this case, the latent factor loadings $\Lambda$ are furthermore modelled as a linear regression of the selected fixed explanatory variables, which in this case was the sentinel plant genotype. Hence, as a summary, our model variants vary in terms of what is included in the matrix $X$ of explanatory variables, namely if sentinel plant genotype is included (variant 2) or not (variant 1), and do we allow the residual associations between viruses to covary among the sentinel genotype (variant 3) or not (variant 2).

We used the default priors of the package 'Hmsc'[108], except that for the parameter $\Lambda$ of species loadings, of the random part of the model. While the HMSC framework is usually not very sensitive to the choices of priors, when data is sufficient, they can be sensitive to the prior chosen for $\Lambda$. The multiplicative gamma process shrinking prior[109] for the species loadings $\Lambda$ has several prior parameters, but out of those, the user is advised to pay attention to the choice of $\alpha$, a vector of two values, which can be used to adjust the level of shrinkage that the prior implies for the matrix $\Omega$ of species associations[106]. Hence, we used two alternative priors. First, we used the default of $\alpha = (50, 50)$, which imposes a lot of shrinkage. We refer to this group of model variants as our main model variants. Second, we used $\alpha = (3,3)$, which imposes much less shrinkage, but as a trade-off, also increases the risk of overfitting.

The model variant comparison approach allows us to examine the relevance of sentinel plant genotype as a predictor of virus community composition (comparison of model variants 1 and 2), as well as to see whether the residual co-occurrences between the viruses differ between the sentinel plant genotypes (variant 3). The comparison of different priors enables us to examine how sensitive our models were for these choices. We compared the model variants in terms of their explanatory and predictive performance, where the first tells us how well the model predicts the data used to fit it, whereas the latter illustrates how well the model predicts independent data which has not been used for model fitting. We calculated the Tjur $R^2$ coefficient of determination, a statistic that has been recommended to be used as a standard measure of explanatory power for binary outcomes[110]. The coefficient is obtained by calculating the mean of the predicted probabilities of presences and absences, and then taking the difference between those two means. Hence, a high coefficient value implies high predicted probabilities for presences and low probabilities for absences. When interpreting it, it is good to note that with sparse data, the probabilities of presence tend to be low in the first place, and thus the Tjur $R^2$ coefficient can remain rather low as well. Nevertheless, if the model is completely uninformative and predicts a 50% probability for both presence and absence, the coefficient value will be zero, thus revealing a poor model fit. For examining explanatory power, we fit the model to the full data set and base our comparison on predictions made for the same data. To examine the predictive power of the model, we conducted a 10-fold cross-validation and compared the model variants based on the same Tjur $R^2$ coefficient as with explanatory power, but calculated from the predictions made to new, unknown host plants. To complement our comparison based on model accuracy, we calculated the widely applicable information criterion (WAIC)[111] for all the variants.

We also conducted a partitioning of the variance explained by the best-performing model variant, to assess how different (groups of) variables are contributing to the overall variance explained by the model at the level of the linear predictor. Finally, we used the best-performing model variant to simulate predicted coinfections profiles.

We implemented our analyses with the R package 'Hmsc' (version 3.0-7[108]). The performance comparison, variance partitioning and predictions were conducted with the tools provided in the package. For a full formal description of the structure of the modelling framework, please see Ovaskainen et al.[104,106], and for the covariate-dependent latent variables used in model variant 3, please see Tikhonov et al.[107]. The analytical pipeline and an R package along with the data used is available in Zenodo (https://doi.org/10.5281/zenodo.4117739). For all the statistical analysis, we used R version 4.0.0[112]. For more details on the statistical analysis, please see Supplementary information (section 'Supplementary information on the joint species distribution modelling').

**Reporting summary.** Further information on research design is available in the Nature Research Reporting Summary linked to this article.

## Data availability

The data supporting our results along with the analytical pipeline implemented as an R package are archived in Zenodo (https://doi.org/10.5281/zenodo.4117739).

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

## Acknowledgements

We thank Mikko Jalo, Pauliina Hyttinen, and Vanja Milenkovic for helping with data collection in the field. We thank Pauliina Hyttinen and Laura Häkkinen for help with RNA-extractions. We thank Krista Raveala for help in cloning and caring for the plants. The Institute of Biotechnology at University of Helsinki is acknowledged for carrying out DNA-extractions. This research was supported by funding from the Academy of Finland (296686) and European Research Council (4100097 RESISTANCE) to A.-L.L., and Luova Doctoral Programme Fellowship to S.S.

## Author contributions

S.S., A.-L.L., and H.S. designed the study. S.S. performed the experiment and data collection. A.N. performed the statistical analysis. S.S., A.N., and A.-L.L. wrote the paper and H.S. contributed substantially with comments on the paper.

## Competing interests

The authors declare no competing interests.
