## [Peer Review File · Nature Communications]

Reviewers' Comments:

Reviewer #1:

Remarks to the Author:

OVERALL

Suvi et al. present an experiment in which they outplant 4 genotypes of a host (*Plantago lanceolata*) into four different natural populations, and then sample to determine whether hosts got infected with any of 5 virus species. They are able to disentangle impacts of host genotype, population (i.e., geographical location), and a few other variables on patterns of occurrence and co-occurrence of the viruses.

Overall, I really like this experiment. It is elegantly designed and tests interesting, novel questions. My primary critique is that readers need more details about the statistical models to be able to interpret some of the results. I am particularly uneasy about inferences drawn from the exclusion of the latent variable for host individual ID – It seems like there ought to be a better way to test for interactions among viruses within hosts. I make a few suggestions for the stats that I think could improve the interpretability of the results. I also found some of the language to be vague, especially in parts of the introduction and discussion, to the point that hindered interpretation and logic (specific places highlighted below).

ABSTRACT

23: Here and elsewhere, consider replacing the phrase “host resistance diversity” as it is a confusing string of nouns. Would “intraspecific variation in disease resistance” be appropriate?

29: This phrasing about JSJM is confusing – it seems to me that this framework lets you test how virus occurrence and co-occurrence patterns differ among host genotypes, host traits, and geographical locations. Is that accurate?

33: What do you mean by “non-random” here?

INTRODUCTION

I find the language in parts of the introduction to be vague to the point where I struggle to follow the authors' reasoning. I have highlighted a few of these sections below. I found some similarly vague statements in the discussion too.

39: “the corner stones of disease” is very vague and bold. What about vectors, immunity, nutrition, stress... And what do you mean by ‘disease’? Do these factors determine infection risk? Spread rates? Community assembly of pathogens? Host mortality?

40: I think you mean multiple infections in an individual host? Not explicitly stated in this first paragraph.

45: “determinants of pathogen communities” is also vague. What about the communities? Determinants of infection risk? Community assembly? Abundance within hosts?

53: What do you mean by “disease abundance” and “disease diversity...” Do you mean pathogen diversity within hosts? Among hosts in a host population?

65: “Host genotype is the key determinant of infection for single pathogens...” statement seems overly bold to me

66: missing word at the end of this sentence?

68: “only recently beginning to gain attention [typo: ettention]...” seems like a stretch to me. This citation (Bolnick et al. 2011 is already almost 10 years old). There is a quite a large literature about eco-evo feedbacks that are only possible because intraspecific variation matters for dynamics at the community and ecosystem scales. There has also been quite a lot of research on disease-community ecology along these lines especially in the *Daphnia-Metschnikowia* system; see work by Spencer Hall, Meg Duffy, and Alex Strauss, among others. Intraspecific variation in host genotypes cascades to impact population and community level outcomes of disease.

72-77: some confusing sentences here, and some grammatical errors

86: here and elsewhere: It is confusing to talk about these as "population level disease drivers," because they are actually at the level of community and ecosystem/environment

89: Somewhere here, you should explain explicitly that you are analyzing the occurrence and co-occurrence patterns of the viruses (more specific than "analyzing the community")

90: "modelling the residual variance with latent variables allows translating these signals into hypothesis of biotic interactions" I may be missing something here, but I don't follow. Can you explain this more biologically, less statistically?

95: "the determinants of pathogen communities" again is very vague. What about the communities? Assembly rules? Coinfection patterns?

97: By "random expectations," do you mean joint probability, i.e., product of each virus's prevalence in singly infected hosts? If so, please make this clear.

98: "affect virus communities" is really vague again. Can this hypothesis be more specific?

98: How is hypothesis 3 different from 1? Is a deviation from random, by definition, either positive or negative?

107: Similar to a comment above: by "population context" do you mean environmental context? Population implies something about the host, which seems misleading here.

METHODS

120: since the early 1990s

141: I'm curious as to why you chose maternal plants that were confirmed to be free of viruses. I don't think this is problematic necessarily, but I wonder you are biasing your results by choosing genotypes that are more resistant than average. I would like to see some discussion of this possibility.

156: This is a very cool experimental design!

166: Assuming that capital A is area, but not explicitly stated (sorry if I missed it). Also be mindful of italics – they are inconsistent here

STATS

207: What does "raw" mean here?

211: Do you mean to say that it calculates expected frequencies by assuming that the species occurrences are independent? This could be a little clearer.

220-231: the model comparison section is confusing. I don't think it provides readers with enough information to be able to interpret your model results. It might be easier to follow if the stat-speak were translated more into biological questions. For example, what are latent factors in this context, and what is a "covariate-dependent latent variable model?"

From this description it is not clear what response variables are being fit by these models (i.e., what multivariate statistic). For example, is it analogous to Jaccard's index in community ecology? Or does it fit each univariate test separately? (It seems like the latter, but this should be clearer in the text). I worry that most readers will not be familiar enough with JSMD to follow this explanation.

Also, it could be useful to explain more philosophically why you are taking the model comparison approach. Why not just fit the full model, and then do a variance partition to figure out relative importance of 'population' vs. host genotype for virus communities? It could also be helpful to state which of your identified research questions 1-5 at the end of the introduction you are addressing with the 'cooccur' package vs. the JSMD framework. Why are you using both of these frameworks, and how do the insights you glean from them complement one another?

I am also quite confused by the latent factors for host individual ID. It seems that you are sometimes including host individual ID (in a sense that seems akin to a random effect to me – apologies if I am misunderstanding) in order to test whether for interactions among viruses within hosts. I could be missing something, but I do not think this approach makes sense, because it tests a very vague question "are individuals different" instead of a specific questions "does infection risk depend on other viruses present". It seems like you could test that hypothesis much more directly by fitting a model that includes 4 additional terms, so occurrence of each virus can depend on whether the host is

infected with each of the other four viruses. So, that would add 4 parameters to each univariate test. It seems to me that you ought to be able to do this, but I apologize if I am missing something here. Regardless, you need to explain more explicitly that the model assumes no interaction among hosts, unless you include the random variable for individual host ID, or preferably if it is feasible, my suggestion above.

RESULTS

253-261: For this section of the results, could you add the expected and observed frequencies for each significant co-occurrence? The methods section also made it seem like they would have p values. These results are fascinating.

266: I think this is the first time you've referred to plant ID as a random effect.

272: What statistics did you use to evaluate the models? AIC or something similar? This model competition needs to be reported.

281-294: It is difficult to tell in this section, which of these differences among host genotypes and viruses are statistically significant. Can you add some p values to explain significance, or at least interpret the model parameters more biologically? Eg., host x was y times more likely to be infected by virus z. If the approach isn't amendable to simple p values, perhaps you could do a set of post-hoc tests for specific comparisons.

300: Important to note the full model R² in this section. Genotype accounts for 47% percent of the explained variation, but the appendix makes it seem like the best model is only explaining ~10% of the variation (which is a fine for ecology – just noting there is a big difference between 47% and 4.7%, and genotype seems to account for 4.7% of the total variation). I'm assuming that I interpret Tjur R² like traditional R², but apologies if I am missing something.

DISCUSSION

317: most of the explained variation (see my comment above about R²)

322: I don't understand how Fig. 2 supports this conclusion. See my questions about the stats section – seems like a more direct test ought to be possible. I don't think JSMD says anything about co-occurrence patterns. In fact, it seems like the current models you have assume that viruses do not interact in host. The fact that the latent variable ('random' effect for individual host ID) did not improve model fit (although not model comparison statistics are presented), does not seem to support the conclusion that "these co-occurrence patterns are influenced more by the local population context and host genotype variation than by direct or indirect interactions among viruses."

329: As explained above, it is quite confusing to refer to abiotic variation as a "population level driver."

339: "several" seems like a major understatement – I'm sure there have been hundreds of studies investigating within-host parasite communities

341: specify, host genotype

353: unclear what you mean here by "population level variation in host resistance"

367: again, I missed the statistical inference to back up the conclusion that the positive and negative co-occurrences were "largely explained by local population context and host genotype." Doesn't the low R² of the best model (~0.1) indicate that most of the co-occurrences were not explained? How does your model test for evidence of interactions among viruses?

376: what do you mean by "induced effects"?

FIGURES & TABLES

Table 1: I would have expected this table to include some model competition statistics (e.g., AIC and evidence ratio or something similar). Can you add some details to help readers evaluate how each model fared?

Table 2: Which model does this table refer to? The previous table introduced 6 models...

Also, although I don't fully understand the JSMD framework, it seems that one level of 'population'

and one level of 'genotype' are arbitrarily set to the intercept, and significance of the terms seems likely to depend on this arbitrary choice. Have you explored whether this is true? Can you justify your choices for the intercept (I noticed that they are not alphabetical, as default in R)?

Figure 2: It's difficult to tell which of the lines the + and - refer to, especially for the interior ones. Can this be made clearer?

Do these statistics take into account the many comparisons that are inherently part of this type of approach? An extra sentence of explanation could help - sorry if I missed it.

Figure 3: Does "empty" mean uninfected?

There are too many colors for the color coding to be particularly helpful. Could you perhaps try collapsing all of the 3-way and 4-way combinations into one color? I suspect that those slivers are really small anyway - their removal could enhance interpretability of the single and 2-way infections.

Figure 4: Probably best to remind readers of the total proportion variance explained by the model. The Tjur R2 in the appendix looks like it is around 0.1.

APPENDIX

I'm having difficulty interpreting Table S3. Can you include the observed and expected frequencies? How do I interpret the 'probability for less' and 'probability for more' columns? Why don't they add up to 1 (what am I missing?)

48: so is model variant 1 the same as the results from the co-occur package?

51: does "host" here mean host individual or host genotype?

Table S4: what does a negative R2 mean?

It seems odd in all of these questions to ask whether you can explain virus occurrence as a function of its prevalence, since occurrence and prevalence seem synonymous. I must be missing something about the way these models work. Can you explain more?

I hope that you find this review helpful.

Signed,

Alex Strauss

Reviewer #2:

Remarks to the Author:

Overview

This manuscript describes an intriguing and original study that investigates the relative importance of host genotype, environmental context, and virus-virus interactions in determining patterns of assembly of virus communities in wild plants. This topic is important in disease ecology because the factors determining patterns of co-infection in the wild are poorly understood, despite the relatively high frequency with which co-infections occur. The study represents a novel application of joint species distribution models (JSDM, applied in the context of hierarchical modeling of species communities, HMSC) to a powerful and unique study system (a well-documented network of *Plantago lanceolata* populations) for which there is extensive long-term ecological data, host genotype information, and previous virus surveys. It thus extends to plant - virus systems HMSC methods previously applied to mammalian parasites (ref. 72 in ms - Dallas, Laine, and Ovaskainen, 2019), here enriched with information about host genotype. The study design is logical and intelligent, and the manuscript is well written and clear in most parts. I appreciated the generally careful tone, and the straightforward acknowledgement of limits (e.g., lines 329-331). The main finding is that both locational context and host genotypes significantly drive virus community assembly in this system, but virus - virus interactions do not. The latter point is counter-intuitive and interesting, as it is frequently assumed that competition and facilitation among viruses are important determinants of co-infection

distribution.

I think this work will be of great interest in the virus and disease ecology communities, as well as among plant community ecologists and others interested in community assembly processes. It has potential to influence these fields by demonstrating new approaches for working with complex microbial data and tackling the important issue of co-infections.

Because of the potential influence of this work, I urge the authors to address three main points that caused some difficulties in reading the manuscript and raised questions. I think all three of these, along with the minor points listed at the end, can be reasonably addressed.

1. Description and discussion of the statistical analysis: JSMD/HMSC and relation to co-occurrence analysis.

A major strength of this study is the novel application of JSMD within HMSC to a plant system with multiple viruses. Thus, it is essential that the HMSC methods and the logic behind them be described as clearly as possible, as a good portion of the intended audience may not yet be familiar with these approaches. In its current format, the discussion of the model terms and logic was not sufficient for this reader. It took a good amount of work to figure out what was going on, and I was never quite sure I'd gotten it right – which meant I felt uncertain about the findings even though I was enthusiastic about the aims. It is my overall sense that the analysis has been handled correctly, although this should be double-checked by an HMSC expert. It is clear to me, however, that the presentation of this section could be strengthened a good deal. In general, what is needed is a clearer explication of the model logic and conceptualization, what the terms represent, which are random and which are fixed, the hierarchical levels, how the performance of the model variants are quantified and compared, and the extent of unexplained variance and what it might represent. This need not be lengthy, but it needs to be more explicit, particularly given the key finding that HMSC uncovered no evidence of virus–virus interactions as manifested by the near-zero contribution of the latent variable Plant ID to residual variance. As an example, I found the methods section in Dallas, T. A., A.-L. Laine and O. Ovaskainen (2019). "Detecting parasite associations within multi-species host and parasite communities." *Proceedings of the Royal Society B: Biological Sciences* 286(1912): 20191109 to be much more clear and effective.

Some additional specific suggestions:

--To gain space for more discussion of the HMSC approach, the discussion of PCR methods can be condensed and details moved to the supplement.

--Throughout, when referring to "genotype" or "population," please specify "plant" or "host," as these terms could equally refer to the viruses. Consider also whether "population" is the best term for the sites at which the sentinel plants were installed. Could not the term 'location' be used? This would make interpretation easier.

-- Lines 81-87 from Supplemental material section on cross-validation and model testing should be moved to main text as they contain essential material for evaluating the soundness of the primary finding.

--Table 1. Include the Tjur R² and Spearman's ρ from Supplemental Table 4 (crucial for evaluating model performance) and at least a short descriptor of what each model variant represents. Were AUC and/or RMSE also used in evaluating model performance? These quantitative comparisons should be noted in Results lines 265-279 because they are critical for the manuscript's main argument.

Related questions in Results:

--Several different confidence intervals are presented (e.g., 95% and 90% in Fig. 2; 95% and 75% in Table 2). Why were these values chosen? I wondered about this in Figure 2, for example, in which the

95% and 90% levels are referred to indistinguishably in the text as 'significant.' I noticed that of the eight co-occurrences marked as significant in Fig. 2, only two are supported at the 95% confidence interval. Hence, if one looked solely at the 95% CI, the difference between the findings of the co-occurrence analysis and the HMSC would be diminished.

--Line 245: Are 49 co-occurrences sufficient for this analysis? I believe they are, but this point should be explicitly addressed.

--Lines 270-277 are among the most important in the manuscript as they describe the key result. Better set-up in the Methods as suggested earlier (e.g., better explanation of terms, model logic, and methods of evaluating models) would them easier to follow.

--Fig. 4 Line 671 refers to "Partitioning of the variance explained by the best performing model variant 3". Elsewhere the text indicates that model 3 was not the best-performing model (lines 269-270: "Model variants 3, 4, and 6 did not differ in their predictive performance.") Rather, Model 3 was one of three equally performing models (Supplementary Table 4, values of Tjur's R² and Spearman's rho). This statement should thus be corrected. Alternatively, if other quantitative criteria beyond Tjur's R² and Spearman's rho were used to evaluate models and these indicate that Model 3 is best, then this additional information should be presented.

Related questions in Discussion:

--So much hinges on interpretation of the HMSC results that discussion of underlying assumptions and short-comings is warranted. As I understand it, an assumption of HMSC (Ovaskainen et al., 2017) is that species – species interactions are captured at the finest scale of sampling (here Plant ID), and for many organisms this makes sense. But is this always true for plant viruses? Although virus-virus interactions most definitely occur within individual plants, they may also occur at other levels, for example within vectors. Since vectors may move over larger areas, these influences have potential to obscure or add complexity to plant-level patterns. The negative finding – virus-virus interactions not important – prompts one to wonder if these interactions could have been wrapped in with other variance or otherwise obscured. More discussion of the HMSC interpretation could resolve this question.

--Lines 282-292 well explain that the finding of the importance of host genotype, as well as the observed pattern of co-occurrences, result from what appears to be increased susceptibility in host genotype 609_19, particularly to capulavirus. The reader thus wonders, was this observational finding empirically tested in a greenhouse experiment? Is there confirming evidence that 609_19 is more disease susceptible? Is it more susceptible to powdery mildew (lines 347 -348)? Anything known about its NLRs (line 350)? If this work hasn't been done, it would be logically described as a key next step.

2. Pathogen frame and focus on resistance

The manuscript is framed in the context of parasites and pathogens, and plant resistance to them, which is one logical framing. But not all viruses are pathogens. Have all five of the viruses studied here been characterized as pathogens, i.e., agents that reduce plant yield or fitness? If so, this point should be made. If not, it would be appropriate to acknowledge this uncertainty and to broaden the framework to consider that these symbionts have unknown influence (+/-/0), so that there may be more mechanisms at work than those of pathogen attack and resistance.

For this reason and others, I find statements such as (line 345-346) "such genotype-level differences are likely to reflect variation in constitutive resistance, such as resistance genes, among the genotypes" not to be entirely convincing. I would like first to be convinced that these viruses are acting as pathogens and then that alternative mechanisms beyond constitutive resistance have been considered and/or evaluated. For example, the genotypes could differ not in resistance per se but in attractiveness to vectors. Likewise, the genotypes could differ in their ability to handle the stress of

having been moved in their pots from the greenhouse to a plastic freezer box in the field. In this case, a stress response that weakened defenses or made plants more attractive could be mistaken for a lack of resistance. If necessary, these issues could be clarified experimentally in the field or greenhouse. But the reader primarily wants to know that they have been considered.

Alternatively, if there is additional existing information that supports the focus on disease resistance per se, it would be valuable for the reader to be shown that. For example, what is already known (i.e., from earlier studies of powdery mildew in these plant populations) about differences in disease resistance among the four genotypes studied here. In the manuscript at present, the only information we have is that (lines 139-141) "The plants are expected to represent four different genotypes (ID:s 609_19, 4_13, 511_14, 2929_6) as their maternal plants originated from distant populations 7-40 kilometres apart." Are the genotypes thus random effects? Or do they represent different levels of resistance in some way? The reader needs this information to interpret the results.

3. "Successful method"

In at least two places in Discussion (e.g., Lines 324-325 and 382 – 385), the text describes this study as demonstrating a "successful method" for studying virus communities in wild hosts. As a reader, I strongly object to this characterization of the work. Yes, absolutely, this is a cool study, the methods have the potential to be powerful, and the results are intriguing. But this paper is in no way a methods paper that demonstrates how well the study of naïve sentinel plants in pots captures the dynamics of virus community assembly within wild field-grown plants. In fact, this essential topic is not discussed at all – it is simply asserted that the study represents a "successful method." The reader wants to know, what are the criteria on which this statement is based? What data demonstrate how well this sentinel pot study captures actual field dynamics?

As a start, for example, is the accumulation of virus infection in 68% of the sentinel plants (line 242) within 7 weeks in the field consistent with prevalence values seen in wild field-grown plants? Is this a plausible number? How do the size and condition of the sentinel plots compare to those in the field? Greenhouse-grown plants frequently are softer and more tender than field-grown individuals, and often differ in growth rates, all of which can influence interactions with vectors and viruses.

More broadly, the reader wants to see discussion of the suite of factors that could lead to divergence between dynamics as captured in the sentinel plants and those that occur in wild plant populations. For example, the use of freezer boxes to isolate potted plants from natural soil may have some benefits (which should be enumerated) but also may confer some disadvantages. I can imagine, for instance, that insect herbivores and their predators will interact with these plants differently than field-grown wild plants, in part because of the box and pot set-up. Crawling insects may be discouraged by the barriers and flying insects may either be attracted or repelled by the visual contrast between plant and box (e.g., winged aphids respond strongly to plant/background contrasts in host choice). In addition, the roots of plants in pots and boxes may be hotter or colder than those growing in the soil, which could alter plant growth and defense processes. Such points need to be considered in any evaluation of how well the sentinel plant study represents true field dynamics.

Minor points

Methods

--Line 125: "geminivirus" should be replaced with "capulavirus".

--Lines 124- 127: Add information about which viruses are RNA species and which are DNA species. While given in the supplement, this information would be helpful here so that readers understand why

both RNA and DNA are extracted for detection.

--Line 141: "free of viruses" should be "free of the target viruses" – text says that plant material was tested with specific primers, not deep-sequenced.

--Line 160-161: Data on potential virus symptoms is collected but not reported further in results or discussion. Was it analyzed?

--Line 161: "significant positive association between capulavirus and closterovirus" is noted for population 433 in the text and Fig. 2, but the statistical support for this association is omitted in supplementary table 3.

--Lines 173-203: Description of PCR methods. These methods are fine but none is particularly unique. So, if space is limited, this material could be shortened and some details moved to supplemental materials, as noted earlier. What would be helpful is more information about the specificity of the primers because this is crucial for the question of the definition of a virus and hence the processes of community assembly. Were amplicons sequenced? What percent identity among sequences of each defined species was evident?

Results

--Measures of prevalence taken twice – week 2 and week 7. Which are presented here?

Discussion

--Lines 321-323 "However, the results of our JSMD modeling indicate that these co-occurrence patterns are influenced more by the local population context and host genotype variation than by direct or indirect interactions among viruses (Fig.2)." Fig. 2 doesn't fully show this. Again, a relatively minor point, but it left me confused because it suggested that there was a single figure that made this point, when in fact one has to consider two different findings together. I had to do a lot of page-flipping to sort this out. To better guide the reader, how about "However, the results of our JSMD modeling (Table 1, expanded as suggested above to include performance measures of models) indicate that the patterns evident in the co-occurrence analysis (Fig. 2) are influenced more by the local population context and host genotype variation than by direct or indirect interactions among viruses."

Tables and Figures

--Fig. 3. Y-axis on right lacks titles. Is 't' the best abbreviation for 'thousand'? Maybe 'k'? Small point, but it confused me at first.

--Fig. 4. The degree of variance explained by host plant size for the closterovirus raises the question of whether the regression coefficient for this was inadvertently omitted in Table 2 or is truly not significant. Worth double-checking.

Best wishes, Carolyn M. Malmstrom

Reviewer #3:

Remarks to the Author:

See attached file.

Review of “Intraspecific host variation plays a key role in virus community assembly”

The authors conduct an experiment to investigate the assembly of pathogen communities as a function of environmental factors and host genotypes. In particular, they estimate the co-occurrence probabilities of pathogens across spatial locations/populations and genotypes. They use joint species distribution models to control for local environment factors as well as host traits.

Overall, the paper is well written. My main comments, given in more detail below, are aimed at the statistical methods, descriptions, and interpretations. While I recognize that this is not a methods paper, since the authors goals are focused around model comparison and model inference to address the objectives outlined on page 5, they should clearly define the model and analytical approaches.

General comments:

1. Be explicit on what the analysis is when using “cooccur.” Is this just a two-way Chi-square test?
2. Write out the JSJM in terms of the random variables, covariates, and parameters along with their distributional assumptions. Define explicitly the number of observations, the covariates, the dependence structure, etc. For example, what are the latent variables? e.g., location/population specific? genotype specific? plant specific? How many are they? Perhaps do this for the full model, noting which pieces are set to 0 for the different model variants. The models are GLMs and GLMMs so it shouldn't take too much space to include in the manuscript. Then, in the results section, use the parameters (e.g., coefficients, variances) in your interpretations in order to illustrate more clearly how these conclusions were reached. For example, how are co-occurrence probabilities obtained? As residual dependencies? How much variation is there among plants, locations/populations, etc.?
3. How is time being accounted for in the model? Is only the final observation time point being modeled? This wasn't made clear in the data section. If multiple observations over time are being included, temporal dependence should be accounted for. Perhaps this is what was meant by plant random effect? Again, writing out the model will help tremendously.

Specific comments:

- Line 167-169: Since host plant size wasn't a significant predictor for any of the pathogens, perhaps remove this as a variable in the model(s). Then this sentence can also be removed. If you keep it in the model, I suggest removing these 13 plants or providing evidence to support that using the averages in place of the missing values has no impact on the results.
- Lines 215-231: Did you consider a genotype by population interaction? If not, why?
- Line 224: What is a “genotype-dependent plant ID-level latent variable”?
- Line 229-230: What is a “covariate dependent latent variable model”?
- Line 269-270: Predictive performance based on what?
- Line 271: Not clear why model 3 is preferred. Random effects to account for random variation between plants, locations/populations, etc. are important, and without including them, results could be biased.

- Line 273: How did you assess for residual co-occurrence?
- Line 281: Did you actually do prediction? Or are these estimates? And how did you estimate co-infections? This paragraph should clearly articulate the results of the model, and how the model was used to make estimates/inference and draw conclusions. Right now, it isn't clear why you fitted a JSDM.
- Line 312: "The host is...", missing a word
- Line 320: JSDM
- Line 320-322: How is this statement justified? Unclear what Figure 2 is trying to show.
- Table 1. Give more details here. How many plant IDs are there? Perhaps even remind the reader about the number of genotypes and wild populations. Host size is a continuous variable and signs of herbivory are categorical, but with how many classes?
- Table 2: Include all of the coefficient estimates. Also, why 75%? I suggest including all, and bolding only those significant at 95% level. Also, which model do these estimates come from? Model 3? Are they consistent with model 6 or any of the others with random effects?
- Figure 3 (lines 666-669): Unclear why MCMC samples are being used to generate these estimates. Can't the estimates of co-infections be obtained directly using parametric expressions? Again, parametric expressions from the model would help.

Response to comments from reviewer 1:

ABSTRACT

23: Here and elsewhere, consider replacing the phrase “host resistance diversity” as it is a confusing string of nouns. Would “intraspecific variation in disease resistance” be appropriate?

Thank you for this suggestion. We agree and have modified the text accordingly on lines 23 and 68.

29: This phrasing about JSMD is confusing – it seems to me that this framework lets you test how virus occurrence and co-occurrence patterns differ among host genotypes, host traits, and geographical locations. Is that accurate?

Yes, this is accurate, and we have changed the wording to be more clear (lines 27-28).

33: What do you mean by “non-random” here?

We simply mean that the occurrences of pathogens are not random, and this results in co-occurrences between the pathogens that would not be predicted based on their frequencies. As we show in our study, this is at least partly explained by the host genotype. Following this comment, we have now modified the wording (lines 31-32), and the topic is further discussed in the Introduction on lines 61-83.

INTRODUCTION

I find the language in parts of the introduction to be vague to the point where I struggle to follow the authors’ reasoning. I have highlighted a few of these sections below. I found some similarly vague statements in the discussion too.

39: “the corner stones of disease” is very vague and bold. What about vectors, immunity, nutrition, stress... And what do you mean by ‘disease’? Do these factors determine infection risk? Spread rates? Community assembly of pathogens? Host mortality?

With “cornerstones of the disease” we are referring to the disease triangle framework (Stevens 1960: *Cultural practices in disease control* in J. G. Horsfall and A. E. Dimond, editors.

Plant pathology, an advanced treatise. 357–429), possibly the longest standing paradigm in plant pathology and crop disease management to describe the conditions required for disease to take place. To clarify this, we have now added the name of the framework to the text (line 39). While this triangle is a tautology (how could the host, pathogen and environment not be essential for disease!), it has been useful in emphasizing that if we ignore or are unable to measure one of the factors, then it is erroneous to ascribe the cause of the disease only to the measured factors. Discussion on how transmission fits into this framework is available for example in Antonovics J. 2017. (*Philos. Trans. R. Soc. Lond. B Biol. Sci* 372 (1719): 20160087). We do not go into such detail here.

We have also revised figure 4 to better convey the disease triangle framework with respect to our results.

By ‘disease’ we mean the standard biological definition of disease: A disease is a particular condition triggered by presence of an infectious agent that negatively affects the structure or function of all or part of an organism. We feel that readers of *Nature Communication* are largely familiar with the concept of disease, and we checked several recent publications, and indeed it had not been further explained. We would prefer to avoid including such explanations as the text will become quite heavy.

40: I think you mean multiple infections in an individual host? Not explicitly stated in this first paragraph.

Precisely. Thank you for pointing this out, we have clarified this in the revision on line 40.

45: “determinants of pathogen communities” is also vague. What about the communities? Determinants of infection risk? Community assembly? Abundance within hosts?

We have clarified this on line 45: “determinants of the assembly and composition of pathogen communities”.

53: What do you mean by “disease abundance” and “disease diversity...” Do you mean pathogen diversity within hosts? Among hosts in a host population?

We have changed our terminology to be clearer, and we now state: “infection load, parasite diversity, and coinfection prevalence across multiple spatial scales” on lines 53-54. These references account for multiple spatial scales such as continents, host populations, and host individuals, and they have measured infection load (quantification of infection e.g. how many parasite individuals are there at a selected location), parasite

diversity (number of pathogen species (species richness and their relative abundances (evenness)), and coinfection (co-occurrence of different parasites).

65: “Host genotype is the key determinant of infection for single pathogens...” statement seems overly bold to me

We have fully revised this sentence to more clearly convey our meaning: “Hosts are resistant against most pathogens species they encounter, and even for pathogens capable of infecting a host species, there is considerable variation among individuals in their susceptibility^{7,44–49}. The effect of this intraspecific variation in disease resistance on the dynamics of individual pathogens is well documented.” (lines 65-67)

66: missing word at the end of this sentence?

We thank the reviewer for noticing and have added the missing word which is “described”. (line 69)

68: “only recently beginning to gain attention [typo: ettention]...” seems like a stretch to me. This citation (Bolnick et al. 2011 is already almost 10 years old). There is a quite a large literature about eco-evo feedbacks that are only possible because intraspecific variation matters for dynamics at the community and ecosystem scales. There has also been quite a lot of research on disease-community ecology along these lines especially in the Daphnia-Metschnikowia system; see work by Spencer Hall, Meg Duffy, and Alex Strauss, among others. Intraspecific variation in host genotypes cascades to impact population and community level outcomes of disease.

Thank you for pointing out the typo — we have now fixed it on line 70. In this paragraph we focus on intraspecific host resistance variation, and we have revised the sentence to clarify this. We have also specified that we mean ‘community assembly and diversity’, our earlier phrasing was too vague. We now state: “However, the importance of intraspecific host disease resistance variation for community assembly and diversity at higher trophic levels is only beginning to gain attention.” (lines 69-70)

We have carefully checked the literature, and we now cite Strauss et al. 2018 (*Funct. Ecol.* 32:1271–1279) on line 69 as an example of the effect of variation in host susceptibility on infection prevalence, and Shoemaker et al. 2019 (*Ecol. Lett.* 22:1115–1125) on line 59 as an example of susceptibility to infection being modified by earlier infection due to vector preference. (The study also examines interspecific variation in host longevity which is outside the scope of our study that focuses on intraspecific variation). Given these

specifications, we feel this statement is correct, but of course it is possible that we may have missed relevant studies and we would be most grateful if these could be pointed out.

72-77: some confusing sentences here, and some grammatical errors

We have fixed the following errors in the language: on line 75 we added “a” before first-arriving pathogens, on line 78 we corrected “has” to “have” as the subject is plural, we moved “are” to a better place (line 82). We also omitted “Each of these mechanisms stemming from” as it was not necessary. In addition, we have specified in this section on lines 81 and 82 that we talk about parasites, not disease.

86: here and elsewhere: It is confusing to talk about these as “population level disease drivers,” because they are actually at the level of community and ecosystem/environment

We thank the reviewer for this comment. We now realize that our communication of the “local population context” was too ambiguous and we have hence clarified it in the introduction (lines 91-93) and thereafter use “population context” when we are referring to the abiotic and biotic factors that affect virus dynamics in the *Plantago lanceolata* populations in the Åland Islands. They may also vary at different spatial scales but the local population is our unit of measurement.

89: Somewhere here, you should explain explicitly that you are analyzing the occurrence and co-occurrence patterns of the viruses (more specific than “analyzing the community”)

We now state explicitly that we are looking into the (co-)occurrence patterns of the viruses. (lines 97-105)

90: “modelling the residual variance with latent variables allows translating these signals into hypothesis of biotic interactions” I may be missing something here, but I don’t follow. Can you explain this more biologically, less statistically?

We modified this part of the introduction and now explain the structure and usage of JSDMs more thoroughly, also paying attention to biological interpretations. On lines 96-105 we now describe the general structure of the modelling framework, how it works in terms of community analysis, and how we use it in this specific case of viruses on plant hosts.

95: “the determinants of pathogen communities” again is very vague. What about the communities? Assembly rules? Coinfection patterns?

We thank the reviewer for pointing this out and have detailed this section by writing “determinants of the assembly and composition of pathogen communities.” (lines 114-115)

97: By “random expectations,” do you mean joint probability, i.e., product of each virus’s prevalence in singly infected hosts? If so, please make this clear.

98: “affect virus communities” is really vague again. Can this hypothesis be more specific?

98: How is hypothesis 3 different from 1? Is a deviation from random, by definition, either positive or negative?

To all three comments above regarding our hypotheses: We have completely re-written this section describing our hypotheses (lines 111-120) to be more to the point, and so that we would address all the issues pointed by the reviewer. Specifically, we now ask whether we find less or more co-occurrences between the viruses than *what would be expected based solely on their marginal frequencies*, and whether the variables affect the virus community *composition*. We have clarified the difference between the hypotheses regarding co-occurrences by emphasizing, that in the latter case we are looking at the *residual* patterns.

107: Similar to a comment above: by “population context” do you mean environmental context? Population implies something about the host, which seems misleading here.

By this we refer to variables that vary at the population level which may include environmental variation, the local disease pool, as well as local vector communities, that all have the potential to affect virus occurrences. We now specify this on lines 91-93.

METHODS

120: since the early 1990s

We have corrected the spelling of this on line 137.

141: I’m curious as to why you chose maternal plants that were confirmed to be free of viruses. I don’t think this is problematic necessarily, but I wonder you are biasing

your results by choosing genotypes that are more resistant than average. I would like to see some discussion of this possibility.

To be able to tease apart variation introduced by the local population context vs. host genotype, it was essential to know that the virus infections we observe were acquired in the field. As the maternal plants were collected as seeds from the natural populations and grown in insect free greenhouse cages, this was merely a check to see whether there is seed borne infection by these viruses. We have clarified this in the text on line 166. Following the comment by reviewer 2, we have edited this to “target viruses” to specify that we confirmed these individuals to be free of the focal viruses before taking them in to the field.

156: This is a very cool experimental design!

We thank the reviewer for these kind words, we are also excited about this approach.

166: Assuming that capital A is area, but not explicitly stated (sorry if I missed it). Also be mindful of italics – they are inconsistent here

Thank you for noting this, we have clarified this on line 192 and corrected the italics.

STATS

207: What does “raw” mean here?

By “raw” we referred to the unmodified presence-absence data of viruses, but we thank the reviewer for pointing out that this wording is not clear and usage of this word unnecessary. Hence, we removed the word. (line 237)

211: Do you mean to say that it calculates expected frequencies by assuming that the species occurrences are independent? This could be a little clearer.

Precisely. We modified the wording for clarity, emphasizing that we are simply comparing observed co-occurrences to expectations based on the frequencies. (lines 241-242)

220-231: the model comparison section is confusing. I don't think it provides readers with enough information to be able to interpret your model results. It might be easier to follow if the stat-speak were translated more into biological questions.

For example, what are latent factors in this context, and what is a “covariate-dependent latent variable model?”

From this description it is not clear what response variables are being fit by these models (i.e., what multivariate statistic). For example, is it analogous to Jaccard’s index in community ecology? Or does it fit each univariate test separately? (It seems like the latter, but this should be clearer in the text). I worry that most readers will not be familiar enough with JSMD to follow this explanation.

Also, it could be useful to explain more philosophically why you are taking the model comparison approach. Why not just fit the full model, and then do a variance partition to figure out relative importance of ‘population’ vs. host genotype for virus communities? It could also be helpful to state which of your identified research questions 1-5 at the end of the introduction you are addressing with the ‘cooccur’ package vs. the JSMD framework. Why are you using both of these frameworks, and how do the insights you glean from them complement one another?

I am also quite confused by the latent factors for host individual ID. It seems that you are sometimes including host individual ID (in a sense that seems akin to a random effect to me – apologies if I am misunderstanding) in order to test whether for interactions among viruses within hosts. I could be missing something, but I do not think this approach makes sense, because it tests a very vague question “are individuals different” instead of a specific questions “does infection risk depend on other viruses present”. It seems like you could test that hypothesis much more directly by fitting a model that includes 4 additional terms, so occurrence of each virus can depend on whether the host is infected with each of the other four viruses. So, that would add 4 parameters to each univariate test. It seems to me that you ought to be able to do this, but I apologize if I am missing something here.

Regardless, you need to explain more explicitly that the model assumes no interaction among hosts, unless you include the random variable for individual host ID, or preferably if it is feasible, my suggestion above.

We thank the reviewer for these in-depth comments. We rewrote this part of our methods completely in order to make our reasoning clearer. We hope that by explaining more thoroughly the model structure as well as how and why the model variant comparison approach was adopted we provide a clearer description of the analysis. We have also simplified our model comparison approach.

We now describe the latent-variables more thoroughly and biologically (lines 99-105 and 246-253). We also describe the whole modelling procedure more thoroughly (lines 244-285). We hope that these modifications highlight the advantages of using this analytical approach where we first analyze the data with an unconstrained, descriptive method of co-occurrence analysis (the cooccur analysis), and then model the community structure, including the co-occurrences, with a constrained model. Furthermore, we hope that the benefits of the chosen JSMD framework, with its hierarchical structure and residual variance modelling, are more clear. Specifically, we want to point out that by using the

latent variable approach, we can elegantly control for the hierarchy of effects, as the interspecific interactions are estimated from the residuals.

We trust that also the philosophy behind the model comparison approach is now clearer, as we simplified the design and rewrote the description (lines 264-280). In short, we want to compare the models so that we can look into the relevance of host genotype as a covariate not only in terms of the explanatory, but also the predictive power of the model, and also see whether the performance improves when we allow the latent variables to covary with host genotype.

RESULTS

253-261: For this section of the results, could you add the expected and observed frequencies for each significant co-occurrence? The methods section also made it seem like they would have p values. These results are fascinating.

We added a sentence stating that the detailed results are included in the supplement (lines 323-325). We had forgotten to state this in the previous version, we apologize for this oversight. The significance levels can be seen in figure 1, and the exact probabilities are presented in the Supplementary table 3, where we now also added the expected and observed values for the significant virus pairs.

266: I think this is the first time you've referred to plant ID as a random effect.

We now elaborate on the latent (random) part of the model in the methods to clarify what we mean by random effects (lines 250-255). We also unified the manuscript so that we talk about 'host plant level' throughout, instead of e.g. 'plant ID'.

272: What statistics did you use to evaluate the models? AIC or something similar? This model competition needs to be reported.

We now describe more clearly in the methods section the performance comparison we conducted (lines 274-280). In summary, we compare the models based on both explanatory and predictive power, measured with the Tjur R^2 coefficient of determination. We also now give the exact values based on which we compare the variants in Table 1.

281-294: It is difficult to tell in this section, which of these differences among host genotypes and viruses are statistically significant. Can you add some p values to explain significance, or at least interpret the model parameters more biologically?

Eg., host x was y times more likely to be infected by virus z. If the approach isn't amendable to simple p values, perhaps you could do a set of post-hoc tests for specific comparisons.

We are using a Bayesian modelling approach which does not provide p-values. We demonstrate the 'significance' of the genotype with our model performance comparison, variance partitioning of the variance explained by the best model, as well as by reporting the posterior means for the regression coefficients along with credible intervals, which inform about the 'significance' of the individual variables, such as the plant genotype (in Table 2). We have revised this section so that we now state also in the text that the genotype had an effect with strong statistical support for many virus species. (lines 340-346)

300: Important to note the full model R2 in this section. Genotype accounts for 47% percent of the explained variation, but the appendix makes it seem like the best model is only explaining ~10% of the variation (which is a fine for ecology – just noting there is a big difference between 47% and 4.7%, and genotype seems to account for 4.7% of the total variation). I'm assuming that I interpret Tjur R2 like traditional R2, but apologies if I am missing something.

This partitioning indeed operates on the variance explained by the model. It is important to note that we are in fact explaining a larger part of the variance, but the coefficient of determination decreases when we make the prediction task more difficult by cross-validating. The variation partitioning is done at the level of the linear predictor by calculating by how much a (group of) explanatory variable(s) contributes to the linear predictor, and we can look at this number in the light of both the explanatory as well as predictive power of our model. We have now clarified this by providing also the explanatory powers of the model variants in Table 1.

DISCUSSION

317: most of the explained variation (see my comment above about R2)

We thank the reviewer for pointing this inaccurate expression. We now changed the wording to be more precise: "we found that most of the model-explained variation in virus occurrences was explained by the local population context and host genotype". (line 388-389)

322: I don't understand how Fig. 2 supports this conclusion. See my questions about the stats section – seems like a more direct test ought to be possible. I don't

think JSMD says anything about co-occurrence patterns. In fact, it seems like the current models you have assume that viruses do not interact in host. The fact that the latent variable ('random' effect for individual host ID) did not improve model fit (although not model comparison statistics are presented), does not seem to support the conclusion that “these co-occurrence patterns are influenced more by the local population context and host genotype variation than by direct or indirect interactions among viruses.”

In the revision as we have simplified the model comparison as well as re-worded the description of this method, we hope to clarify the reasoning behind our analysis of co-occurrences: We see virus occurrences and co-occurrences in the data, we fit the model and explain those patterns with the effect of host genotype and traits, and local population context, and after we control for these fixed effects, we no longer detect significant co-occurrence and after we control for these fixed effects, we no longer detect significant co-occurrence patterns in the residuals that would be indicative of interactions among the viruses. In other words, our model results tell us that the co-occurrence detected in the observational data are driven by plant traits and local population context rather than virus-virus interactions.

329: As explained above, it is quite confusing to refer to abiotic variation as a “population level driver.”

As in our response above, we thank the reviewer for their comment which demonstrates to us that our communication of the “local population context” was too ambiguous and we have hence clarified it in the introduction (lines 91-93) and further keep referring to “population context” when we are referring to the abiotic and biotic factors that affect virus dynamics in the *Plantago lanceolata* populations in the Åland Islands.

339: “several” seems like a major understatement – I’m sure there have been hundreds of studies investigating within-host parasite communities

We agree that “several” is indeed an understatement and we have changed this to “multiple studies” on line 415. The main message we want to deliver with this sentence is that while there have been hundreds of studies on the topic, there have not been previous studies that would have attempted to identify the effect of host genotype on parasite communities as we do here.

341: specify, host genotype

Throughout the text, we have now specified “host genotype” or “host plant genotype”.

353: unclear what you mean here by “population level variation in host resistance”

Following the previous comments concerning “host disease diversity”, we have modified this sentence on line 431-432 to the following: “...-based on our findings, intraspecific variation in disease resistance in a host population may play a large...-”

367: again, I missed the statistical inference to back up the conclusion that the positive and negative co-occurrences were “largely explained by local population context and host genotype.” Doesn’t the low R² of the best model (~0.1) indicate that most of the co-occurrences were not explained? How does your model test for evidence of interactions among viruses?

Here, like above, we trust that our simplified modelling pipeline along with reporting of both predictive and explanatory powers of the models as well as change of wording more clearly conveys our reasoning: By looking at the co-occurrence patterns in the data we can see that there are more (or less, respectively) co-occurrences of viruses in the host plants than would be expected solely based on their frequencies. Then, as we apply a JSDM, we see that those co-occurrences are explained by mostly the host plant genotype along with local plant population context, with no signals of residual co-occurrences left. Hence, we conclude that the patterns we detected in the data were due to the effects of the genotype and the local population.

376: what do you mean by “induced effects”?

We now clarify that we mean “the effects of induced immunity triggered by a first arriving pathogen” on line 459-460.

FIGURES & TABLES

Table 1: I would have expected this table to include some model competition statistics (e.g., AIC and evidence ratio or something similar). Can you add some details to help readers evaluate how each model fared?

We now include both explanatory and predictive power for the model variants in this table. We also now added a paragraph in the ‘Comparison of model performance’ section of the Supplement where we give more background to this method.

Table 2: Which model does this table refer to? The previous table introduced 6 models...

Thank you for pointing this out, we now clarify that the regression coefficients are for the best-performing model, i.e. variant 2, out of the total of three candidates we now have in our simplified design.

Also, although I don't fully understand the JSDM framework, it seems that one level of 'population' and one level of 'genotype' are arbitrarily set to the intercept, and significance of the terms seems likely to depend on this arbitrary choice. Have you explored whether this is true? Can you justify your choices for the intercept (I noticed that they are not alphabetical, as default in R)?

Yes, the choice of the intercept levels is arbitrary. We do not base our results merely on the 'significance' of the posterior estimates of the genotypes nor populations, but on that our best models include both of these variables as well as on that when looking at the variance partitioning, we see these two explaining the majority of variance captured by our model. Our discussion related to the effects of specific genotypes is very exploratory and we are not attempting to draw any strong conclusions of the effects of specific genotypes, but rather conclude that overall, they have an effect.

Figure 2: It's difficult to tell which of the lines the + and – refer to, especially for the interior ones. Can this be made clearer?

We increased both the font size as well as contrast of the signs.

Do these statistics take into account the many comparisons that are inherently part of this type of approach? An extra sentence of explanation could help – sorry if I missed it.

This method does not make any corrections to account for multiple testing. However, our approach is rather descriptive, as we merely state that we see some co-occurrences between species, and our analysis method provides the probabilities for them. We do not draw conclusions from individual pairs of species, but from the overall finding of co-occurrences, as seen already from the data. Because we have to make the cut somewhere, we only show the pairs with >90% probability, but we provide the exact probabilities for all pairs in the Supplement (Supplementary table 3), along with the expected and observed values. We also now give the expected and observed values in the main text.

Figure 3: Does “empty” mean uninfected?

There are too many colors for the color coding to be particularly helpful. Could you perhaps try collapsing all of the 3-way and 4-way combinations into one color? I suspect that those slivers are really small anyway – their removal could enhance interpretability of the single and 2-way infections.

We thank the reviewer for this suggestion, we have changed the legend so that we use ‘Uninfected’ instead of ‘Empty’. We understand that the multicombinations are difficult to tell apart, nevertheless, we feel that the figure is more informative as it is, because if we would pool them, it would not come through that there are these single cases of combinations that the model predicts. In the current form the main types of (co-)infection can be separated and the ones that cannot still provide information we would not want to leave out.

Figure 4: Probably best to remind readers of the total proportion variance explained by the model. The Tjur R2 in the appendix looks like it is around 0.1.

We thank the reviewer for pointing this out. In the legend we specify that the figure depicts “Partitioning of the variance explained by the model- variant 2”. All model performance measures are presented in Table 1.

APPENDIX

I’m having difficulty interpreting Table S3. Can you include the observed and expected frequencies? How do I interpret the ‘probability for less’ and ‘probability for more’ columns? Why don’t they add up to 1 (what am I missing?)

We added more explanation to this section of both the Methods and the Supplement. We also now provide both the expected and observed values, and we have revised the figure legend to be more informative. The Supplementary table 3 shows probabilities for pairs of species co-occurring more (or less, respectively) than the observed number of co-occurrences if these two species were distributed randomly, independently of one another. In the figure, we indicate the pairs for which the probability of more (or less) co-occurrences is less than 0.1, but the observed number of co-occurrences is greater (or lesser, respectively).

48: so is model variant 1 the same as the results from the co-occur package?

We removed this variant as redundant in revision aiming to simplify the logic in our model comparison approach and to better illustrate the main findings.

51: does “host” here mean host individual or host genotype?

Host individual. We now refer to ‘host plant individual’ throughout the main and supplementary text.

Table S4: what does a negative R² mean?

Due to the way the Tjur R² is calculated it can result in negative values. The coefficient is obtained by summing over probabilities of presences of the observation matrix cells for which the true value is 1, and probabilities for absences of the cells for which the true value is 0, and dividing this by the total number of cells. Hence, if the predicted probabilities are high for presences and low for absences, the coefficient gets a value close to 1. With sparse data, where the occurrence probabilities are in general low due to species low prevalence, the value can easily remain low, even if the predicted probabilities for true absences are generally lower than the ones predicted for true presences. But when there is a great deal of stochasticity in the data, even negative coefficient values are reached, because this classification fails. Please see Tjur 2009 (*Am.stat.* 63:366-372) for more details.

It seems odd in all of these questions to ask whether you can explain virus occurrence as a function of its prevalence, since occurrence and prevalence seem synonymous. I must be missing something about the way these models work. Can you explain more?

We agree, that the previous wording could be confusing to the reader. The usage of prevalence is easiest explained through our intercept-only model variant (which is now removed from the revised selection of model variants): if we model the occurrence *patterns* as a function of just the intercept, we basically just try to explain the occurrences of the species by a constant describing its commonness — or, as we said, prevalence. To clarify this, we have rewritten the model variant descriptions, and we trust that they are now more informative.

Response to comments from reviewer 2:

1. Description and discussion of the statistical analysis: JSJM/HMSC and relation to co-occurrence analysis.

A major strength of this study is the novel application of JSJM within HMSC to a plant system with multiple viruses. Thus, it is essential that the HMSC methods and the logic behind them be described as clearly as possible, as a good portion of the intended audience may not yet be familiar with these approaches. In its current format, the discussion of the model terms and logic was not sufficient for this reader. It took a good amount of work to figure out what was going on, and I was never quite sure I'd gotten it right – which meant I felt uncertain about the findings even though I was enthusiastic about the aims. It is my overall sense that the analysis has been handled correctly, although this should be double-checked by an HMSC expert. It is clear to me, however, that the presentation of this section could be strengthened a good deal. In general, what is needed is a clearer explication of the model logic and conceptualization, what the terms represent, which are random and which are fixed, the hierarchical levels, how the performance of the model variants are quantified and compared, and the extent of unexplained variance and what it might represent. This need not be lengthy, but it needs to be more explicit, particularly given the key finding that HMSC uncovered no evidence of virus–virus interactions as manifested by the near-zero contribution of the latent variable Plant ID to residual variance. As an example, I found the methods section in Dallas, T. A., A.-L. Laine and O. Ovaskainen (2019). "Detecting parasite associations within multi-species host and parasite communities." *Proceedings of the Royal Society B: Biological Sciences* 286(1912): 20191109 to be much more clear and effective. Some additional specific suggestions:

--To gain space for more discussion of the HMSC approach, the discussion of PCR methods can be condensed and details moved to the supplement.

We thank the reviewer for this comment. We have now considerably expanded the description of the modelling framework both in the introduction (lines 96-106) as well as in the methods (244-285). We also simplified the modelling pipeline by dropping redundant model variants. We now explain the model comparison procedure in more detail and give the predictive and explanatory performance results in Table 1.

--Throughout, when referring to “genotype” or “population,” please specify “plant” or “host,” as these terms could equally refer to the viruses. Consider also whether “population” is the best term for the sites at which the sentinel plants were installed. Could not the term ‘location’ be used? This would make interpretation easier

We thank the reviewer for pointing this out and have specified throughout “host” or “plant”, and further so that when referring to our research plant populations, we say “local population context” and when referring to general phenomenon, we say “host population”.

We are using “population” instead of location as these are small individual populations of *Plantago lanceolata* used for decades of research in metapopulation studies carried out in this system with established population ID:s (Hanski 1999 *Metapopulation Ecology*, Oxford University Press; Ojanen et al. 2013 *Ecol.Evol.* 3: 3713–3737). We further lay out our research questions and hypothesis so that we use “local population context” as the variable from which we tease apart our genotype effect from (lines 113, 116).

-- Lines 81-87 from Supplemental material section on cross-validation and model testing should be moved to main text as they contain essential material for evaluating the soundness of the primary finding.

--Table 1. Include the Tjur R² and Spearman’s ρ from Supplemental Table 4 (crucial for evaluating model performance) and at least a short descriptor of what each model variant represents. Were AUC and/or RMSE also used in evaluating model performance? These quantitative comparisons should be noted in Results lines 265-279 because they are critical for the manuscript’s main argument.

We thank the reviewer for these comments. We moved the description of cross-validation to the methods (lines 274-278), and we now provide both the explanatory and predictive power values in Table 1 in the main text. We report the host plant individual level results in the main text and base our model comparison to those. We provide the genotype-by-population-level performance results in the Supplementary table 4, because we feel that this is indeed supplementary information.

Related questions in Results:

--Several different confidence intervals are presented (e.g., 95% and 90% in Fig. 2; 95% and 75% in Table 2). Why were these values chosen? I wondered about this in Figure 2, for example, in which the 95% and 90% levels are referred to indistinguishably in the text as ‘significant.’ I noticed that of the eight co-occurrences marked as significant in Fig. 2, only two are supported at the 95% confidence interval. Hence, if one looked solely at the 95% CI, the difference between the findings of the co-occurrence analysis and the HMSC would be diminished.

We agree that our choice of thresholds for results with strong statistical support are not consistent, and we have revised the text to clarify why this is the case. This was partly due to the selection of methods. Our modelling framework is Bayesian, where there are no strict conventions regarding significance thresholds. To provide as complete view as possible of our results, we display results for different levels of statistical support. We now state that the residual co-occurrence patterns were completely random, even when examined using lower significance thresholds than e.g. 95% . We also changed the

thresholds for significance so that we use 90% consistently throughout the manuscript (i.e. Figure 2 and Table 2).

--Line 245: Are 49 co-occurrences sufficient for this analysis? I believe they are, but this point should be explicitly addressed.

We agree that this is relevant to clarify, and we now clearly state already in the Introduction that our JSDM method of choice is robust against sparse data (line 99) and we also provide the exact numbers for observed and expected co-occurrences for the species pairs for which we detected significantly more or less co-occurrences in Supplementary Table 3.

--Lines 270-277 are among the most important in the manuscript as they describe the key result. Better set-up in the Methods as suggested earlier (e.g., better explanation of terms, model logic, and methods of evaluating models) would make them easier to follow.

--Fig. 4 Line 671 refers to “Partitioning of the variance explained by the best performing model variant 3”. Elsewhere the text indicates that model 3 was not the best-performing model (lines 269-270: “Model variants 3, 4, and 6 did not differ in their predictive performance.”) Rather, Model 3 was one of three equally performing models (Supplementary Table 4, values of Tjur’s R² and Spearman’s rho). This statement should thus be corrected. Alternatively, if other quantitative criteria beyond Tjur’s R² and Spearman’s rho were used to evaluate models and these indicate that Model 3 is best, then this additional information should be presented.

We thank the reviewer for these insightful comments. We fully agree, and we have now simplified the model variant comparison, and completely rewritten the methods section regarding the modelling procedure. We now also provide the explanatory power of the model variants, and we rewrote the key section of the results in a clearer manner (lines 334-338).

Related questions in Discussion:

--So much hinges on interpretation of the HMSC results that discussion of underlying assumptions and short-comings is warranted. As I understand it, an assumption of HMSC (Ovaskainen et al., 2017) is that species – species interactions are captured at the finest scale of sampling (here Plant ID), and for many organisms this makes sense. But is this always true for plant viruses? Although virus-virus interactions most definitely occur within individual plants, they may also occur at other levels, for example within vectors. Since vectors may move over larger areas, these influences have potential to obscure or add complexity to plant-level patterns.

The negative finding – virus-virus interactions not important – prompts one to wonder if these interactions could have been wrapped in with other variance or otherwise obscured. More discussion of the HMSC interpretation could resolve this question.

This is a very good point, and we appreciate this insightful comment. It is absolutely true that it is solely on the shoulders of the modeler to decide what is the relevant level for observing the residual co-occurrences. Biologically, we expect both direct virus-virus interactions and those mediated by the host to take place at the scale of a host individual, but indeed arrival probabilities are strongly affected by vector dynamics at greater spatial scales. In our data we expect the ‘local population context’ to capture some of this variation — and we now specify this on lines 91-93 and 180-181 — but unfortunately we do not have the sufficient data to look into this. At the plant level our sample size is 320, while at the population level our sample size is four. In addition, we do not have the population-level abiotic variables to tease apart their effect from e.g. the effect of vectors. By including population as a fixed effect, we hope to account for at least the most clear-cut processes operating at this level. By accounting for both population and plant-level variables we trust that we can interpret the residual patterns at the host plant individual level.

--Lines 282-292 well explain that the finding of the importance of host genotype, as well as the observed pattern of co-occurrences, result from what appears to be increased susceptibility in host genotype 609_19, particularly to capulavirus. The reader thus wonders, was this observational finding empirically tested in a greenhouse experiment? Is there confirming evidence that 609_19 is more disease susceptible? Is it more susceptible to powdery mildew (lines 347 -348)? Anything known about its NLRs (line 350)? If this work hasn't been done, it would be logically described as a key next step.

We currently don't have data from lab inoculations or NLR -level resistance against viruses in *P. lanceolata*, the current information on among plant level variation comes from field surveys, this experiment, and ongoing common garden trials. However, these genotypes are known to differ in their resistance against the powdery mildew which we know from extensive lab strain maintenance data. We've now clarified this on lines 193-195. We also now identify this as an interesting future research direction in the Discussion on lines 429-430.

2. Pathogen frame and focus on resistance

The manuscript is framed in the context of parasites and pathogens, and plant resistance to them, which is one logical framing. But not all viruses are pathogens. Have all five of the viruses studied here been characterized as pathogens, i.e.,

agents that reduce plant yield or fitness? If so, this point should be made. If not, it would be appropriate to acknowledge this uncertainty and to broaden the framework to consider that these symbionts have unknown influence (+/-/0), so that there may be more mechanisms at work than those of pathogen attack and resistance.

This is an important point, and we had not appropriately acknowledged the complex ecological roles that viruses play in natural plant populations. Moreover, we had not included some relevant information based on which we expect these viruses to act as potential pathogens, at least under some circumstances. First, the viruses were initially identified from *P. lanceolata* in Åland by sequencing plant small RNAs (Susi et al. 2019 *PeerJ* 7:e6140). Plants use RNA-silencing mechanism and produce short interfering RNA (SiRNA) molecules in a defense response against viral infection (Baulcombe 2004 *Nature* 431: 356–363). Hence, these viruses trigger an active defense response in *P. lanceolata*. Also, although not directly demonstrating their pathogenic nature, Susi et al 2019 (*PeerJ* 7:e6140) found that plants with virotic symptoms (necrotic spots/yellow color) are more likely to carry a virus infection. We now clarify these points on lines 144-150. We have done a common garden experiment where we investigate the ecological interactions between *P. lanceolata* and these viruses but we do not have the results yet.

To better acknowledge that viruses move across a mutualist-antagonist continuum, we have added a section describing the complex ecological roles of plant viruses both in the Introduction (lines 86-89) and the Discussion (lines 480-483), where we also acknowledge that other traits could be involved, and that future studies should explore in more detail the ecological outcomes and molecular underpinning of these interactions (lines 435-437). We also now highlight the need for further research on viruses in natural communities (lines 474-476).

For this reason and others, I find statements such as (line 345-346) “such genotype-level differences are likely to reflect variation in constitutive resistance, such as resistance genes, among the genotypes” not to be entirely convincing. I would like first to be convinced that these viruses are acting as pathogens and then that alternative mechanisms beyond constitutive resistance have been considered and/or evaluated. For example, the genotypes could differ not in resistance per se but in attractiveness to vectors. Likewise, the genotypes could differ in their ability to handle the stress of having been moved in their pots from the greenhouse to a plastic freezer box in the field. In this case, a stress response that weakened defenses or made plants more attractive could be mistaken for a lack of resistance. If necessary, these issues could be clarified experimentally in the field or greenhouse. But the reader primarily wants to know that they have been considered.

We agree with the reviewer and as also specified in our previous response, we now present resistance as a possible underlying mechanism, supported in part by our previous

knowledge which suggests these viruses to have a pathogenic role under some circumstances. We now also acknowledge more clearly that these plants could differ in other attributes, and we highlight the possibility that attractiveness to vectors could also generate these differences. The plants did not display signs of stress (typically seen as discoloration of leaves) but of course we cannot exclude this possibility. We now state that other traits than resistance could be involved, but avoid further speculation on what those traits might be. (lines 435-437)

Alternatively, if there is additional existing information that supports the focus on disease resistance per se, it would be valuable for the reader to be shown that. For example, what is already known (i.e., from earlier studies of powdery mildew in these plant populations) about differences in disease resistance among the four genotypes studied here. In the manuscript at present, the only information we have is that (lines 139-141) “The plants are expected to represent four different genotypes (ID:s 609_19, 4_13, 511_14, 2929_6) as their maternal plants originated from distant populations 7-40 kilometres apart.” Are the genotypes thus random effects? Or do they represent different levels of resistance in some way? The reader needs this information to interpret the results.

As specified in our previous responses, we have updated the manuscript with critical information that we had failed to include in the earlier version: These viruses were initially found from *P. lanceolata* populations in Ålanad based on a plant defense response, and they associate with symptoms in the natural populations (lines 144-145 and 148-150). We also now state that these genotypes are known to vary in their resistance against the powdery mildew (lines 164-165). However, we would prefer not to make comparisons between mildew resistance and our current results – to establish such links we would need a much larger sample size. Our intention here was merely to test whether genotypes (and we can assume these to be distinct genotypes based on their mildew resistance) originating from the same limited geographic area accumulate different virus communities. The fact that they do serves as a valuable starting point for future studies to investigate this in more detail. We now state this on lines 429-430.

3. “Successful method”

In at least two places in Discussion (e.g., Lines 324-325 and 382 – 385), the text describes this study as demonstrating a “successful method” for studying virus communities in wild hosts. As a reader, I strongly object to this characterization of the work. Yes, absolutely, this is a cool study, the methods have the potential to be powerful, and the results are intriguing. But this paper is in no way a methods paper that demonstrates how well the study of naïve sentinel plants in pots captures the dynamics of virus community assembly within wild field-grown plants. In fact, this essential topic is not discussed at all – it is simply asserted that the study represents a “successful method.” The reader wants to know, what are the criteria

on which this statement is based? What data demonstrate how well this sentinel pot study captures actual field dynamics?

As a start, for example, is the accumulation of virus infection in 68% of the sentinel plants (line 242) within 7 weeks in the field consistent with prevalence values seen in wild field-grown plants? Is this a plausible number? How do the size and condition of the sentinel plots compare to those in the field? Greenhouse-grown plants frequently are softer and more tender than field-grown individuals, and often differ in growth rates, all of which can influence interactions with vectors and viruses.

More broadly, the reader wants to see discussion of the suite of factors that could lead to divergence between dynamics as captured in the sentinel plants and those that occur in wild plant populations. For example, the use of freezer boxes to isolate potted plants from natural soil may have some benefits (which should be enumerated) but also may confer some disadvantages. I can imagine, for instance, that insect herbivores and their predators will interact with these plants differently than field-grown wild plants, in part because of the box and pot set-up. Crawling insects may be discouraged by the barriers and flying insects may either be attracted or repelled by the visual contrast between plant and box (e.g., winged aphids respond strongly to plant/background contrasts in host choice). In addition, the roots of plants in pots and boxes may be hotter or colder than those growing in the soil, which could alter plant growth and defense processes. Such points need to be considered in any evaluation of how well the sentinel plant study represents true field dynamics.

We thank the reviewer for pointing out this unnecessarily strong emphasis on methods that lacks a critical examination of its possible limitations. We agree with the reviewer that this is not a methods paper per se, in our excitement about the results that our approach generated we had placed too much emphasis on this aspect. We have revised the text to avoid this. Instead, we highlight how our results may differ from the dynamics in natural plants, and the need for further research in the field of virus ecology (lines 466-479). We do, however, want to emphasize that teasing apart the genotype from other factors is difficult, and such set-up has not been previously utilized in this context. We now discuss this idea in a more fitting manner in discussion (400-403, 485-487).

Indeed, our plants do not perfectly resemble the wild plants for the reasons listed by the reviewer, and we now address the limitations and opportunities by this sentinel host approach in the Discussion on lines 466-470. We aim to keep this brief to indeed not convey our study as a methods paper. Virus prevalences are highly variable in both space and in the Åland Island *P. lanceolata* populations: Susi et al 2019 (*PeerJ* 7:e6140) found prevalences ranging from 0 to 64 % among *P. lanceolata* populations. The prevalences we find in this sampling are higher than found in the Susi et al. 2019 but lower than in a small-RNA sequencing data set spanning 20 *P. lanceolata* populations and 400 individual plants

where 71% of plants were infected by at least one virus, and among population prevalences varied between 0-90%. We now discuss this on lines 472-474.

Minor points

Methods

--Line 125: “geminivirus” should be replaced with “capulavirus”.

We thank the reviewer for spotting this error and have corrected it on line 141.

--Lines 124- 127: Add information about which viruses are RNA species and which are DNA species. While given in the supplement, this information would be helpful here so that readers understand why both RNA and DNA are extracted for detection.

Thank you for this suggestion, we have added this information. (lines 141-144)

--Line 141: “free of viruses” should be “free of the target viruses” – text says that plant material was tested with specific primers, not deep-sequenced.

Thank you for noticing, we have edited the text accordingly on line 166.

--Line 160-161: Data on potential virus symptoms is collected but not reported further in results or discussion. Was it analyzed?

Indeed, data collection included virus symptoms but these data were not included in these analysis as it would be circular to explain virus communities with symptoms that the virus communities may cause. We have hence removed this from the manuscript.

--Line 161: “significant positive association between capulavirus and closterovirus” is noted for population 433 in the text and Fig. 2, but the statistical support for this association is omitted in supplementary table 3.

We have corrected the table, thank you for noticing.

--Lines 173-203: Description of PCR methods. These methods are fine but none is particularly unique. So, if space is limited, this material could be shortened and

some details moved to supplemental materials, as noted earlier. What would be helpful is more information about the specificity of the primers because this is crucial for the question of the definition of a virus and hence the processes of community assembly. Were amplicons sequenced? What percent identity among sequences of each defined species was evident?

Thank you for the suggestion. As space is not limited, we have left these methods into the text for reproducibility. The primers have been design using small-RNA sequencing data collected from *P. lanceolata* populations in the Åland Islands, and are specific for the strains that occur in this system. In the present study, we only use the primers developed for detection of these strains, and have not explored the sequence similarities by sequencing the amplicons. The characterization of these viruses, for which no reference sequence was found in existing databases, and the development of these detection primers is presented in Susi et al 2019 (*PeerJ* 7:e6140), and Susi et al 2017 (*Arch.Virol.* 162: 2041–2045).

Results

--Measures of prevalence taken twice – week 2 and week 7. Which are presented here?

We thank the reviewer for pointing this out, and this is now clarified in the beginning of 'Statistical analysis' in the Methods: we pooled the data of these two time points. (lines 234-235)

Discussion

--Lines 321-323 "However, the results of our JSMD modeling indicate that these co-occurrence patterns are influenced more by the local population context and host genotype variation than by direct or indirect interactions among viruses (Fig.2)." Fig. 2 doesn't fully show this. Again, a relatively minor point, but it left me confused because it suggested that there was a single figure that made this point, when in fact one has to consider two different findings together. I had to do a lot of page-flipping to sort this out. To better guide the reader, how about "However, the results of our JSMD modeling (Table 1, expanded as suggested above to include performance measures of models) indicate that the patterns evident in the co-occurrence analysis (Fig. 2) are influenced more by the local population context and host genotype variation than by direct or indirect interactions among viruses."

We fully agree that figure 2 only displays part of this result. We thank the reviewer for providing an excellent suggestion, and we have rewritten this part of the discussion accordingly. (lines 394-396)

Tables and Figures

--Fig. 3. Y-axis on right lacks titles. Is 't' the best abbreviation for 'thousand'? Maybe 'k'? Small point, but it confused me at first.

We agree and have modified the axis labels from 't' to 'k'.

--Fig. 4. The degree of variance explained by host plant size for the closterovirus raises the question of whether the regression coefficient for this was inadvertently omitted in Table 2 or is truly not significant. Worth double-checking.

We now give all the posterior mean estimates in Table 2, with the ones with strong statistical support in bold.

Best wishes, Carolyn M. Malmstrom

Response to comments from the reviewer 3:

General comments:

1. Be explicit on what the analysis is when using "cooccur." Is this just a two-way Chi-square test?

We expanded the explanation of the method in the beginning of the 'Statistical analysis' section (lines 235-242). The applied model gives us the probabilities that two species occur at a frequency greater (or less) than the observed frequency if the two species were distributed independently of one another. The method relies on combinatorics, does not require any randomizations, and is thus purely analytical and distribution-free.

2. Write out the JSMD in terms of the random variables, covariates, and parameters along with their distributional assumptions. Define explicitly the number of observations, the co-variates, the dependence structure, etc. For example, what are the latent variables? e.g., location/population specific?

genotype specific? plant specific? How many are they? Perhaps do this for the full model, noting which pieces are set to 0 for the different model variants. The models are GLMs and GLMMs so it shouldn't take too much space to include in the manuscript. Then, in the results section, use the parameters (e.g., coefficients, variances) in your interpretations in order to illustrate more clearly how these conclusions were reached. For example, how are co-occurrence probabilities obtained? As residual dependencies? How much variation is there among plants, locations/populations, etc.?

We thank the reviewer for pointing out that the modelling framework should be explained in more detail. We have now included a more thorough explanation of the model, with clear references to the publications explaining the exact formulation of the model (lines 255-262), but we feel that similar level of detail for an already published method is perhaps slightly beyond the scope of the current study. Regarding all distributions, we are relying on the defaults of the method, hence the detailed exact formal descriptions are found in the provided references, and we added now also a general formal description in the Supplement.

3. How is time being accounted for in the model? Is only the final observation time point being modeled? This wasn't made clear in the data section. If multiple observations over time are being included, temporal dependence should be accounted for. Perhaps this is what was meant by plant random effect? Again, writing out the model will help tremendously.

We apologise for overlooking this matter. We explain now in the beginning of statistical analysis in the methods section that the virus occurrences were pooled over the time points before the analyses. (line 234-235)

Specific comments:

•Line 167-169: Since host plant size wasn't a significant predictor for any of the pathogens, perhaps remove this as a variable in the model(s). Then this sentence can also be removed. If you keep it in the model, I suggest removing these 13 plants or providing evidence to support that using the averages in place of the missing values has no impact on the results.

Our logic with the covariates is, that we have a ‘basic set’ of variables, namely, the local environment, herbivory and host plant size, which we assume to influence the pathogen community assembly. To our knowledge, there are not many studies showing the effects of host genotype, and hence we are looking into its effects by comparing the baseline model and two different ways of including the host genotype. In our opinion removing this variable would violate the representation we have for the system, which is based on theory and previous studies. Even though the plant size does not result in parameter estimates with strong statistical support, it does not mean that this variable would not contribute to the model at all, as seen from the variance partitioning (Figure 4).

•Lines 215-231: Did you consider a genotype by population interaction? If not, why?

Given the limited amount of data, the inclusion of interaction terms in the set of explanatory variables would increase the risk of overfitting. Even the inclusion of the main effects results in 6 explanatory variables, in addition to the plant size and signs of herbivory.

•Line 224: What is a “genotype-dependent plant ID-level latent variable”?

•Line 229-230: What is a “covariate dependent latent variable model”?

In a covariate- (or in this case genotype-) dependent latent variable model we allow the latent variables to covary with a selected fixed effect (in this case, the host genotype). We now explain these more clearly in the methods. (lines 244-272)

•Line 269-270: Predictive performance based on what?

The predictive performance was based on Tjur R^2 values calculated for each variant. This is now explained more clearly in the methods (lines 274-283) and results are provided in Table 1.

•Line 271: Not clear why model 3 is preferred. Random effects to account for random variation between plants, locations/populations, etc. are important, and without including them, results could be biased.1

We thank the Reviewer for this suggestion. We have modified the model variant comparison and justification for selecting the best one (lines 330-338), and we now keep the random effects in the final model.

•Line 273: How did you assess for residual co-occurrence?

We used a latent variable approach, which allows us to model the residual variance after accounting for the fixed effects. We can transform the latent factor loading into a matrix of residual correlations between species, for which we can also calculate credible intervals and thus assess their statistical support. We now explain this more clearly in the methods. (lines 244-272)

•Line 281: Did you actually do prediction? Or are these estimates? And how did you estimate co-infections? This paragraph should clearly articulate the results of the model, and how the model was used to make estimates/inference and draw conclusions. Right now, it isn't clear why you fitted a JSJM.

These results are based on cross-validation predictions, i.e. we divided the data randomly in 10 parts and made the predictions for one-fold (1/10 data) based on a model fitted with the rest of the data (9/10). We now explain how these predictions were made already in the methods. (lines 275-278)

•Line 312: "The host is...", missing a word

Corrected. (line 383)

•Line 320: JSJM

Corrected. (line 394)

•Line 320-322: How is this statement justified? Unclear what Figure 2 is trying to show.

We modified this sentence: "the results of our JSJM modeling (Table 1) indicate that the patterns evident in the co-occurrence analysis (Fig. 2) are influenced more by the local population context and host genotype variation than by direct or indirect biotic interactions among the viruses" (lines 394-396)

•Table 1. Give more details here. How many plant IDs are there? Perhaps even remind the reader about the number of genotypes and wild populations. Host size is a continuous variable and signs of herbivory are categorial, but with how many classes?

We decided to keep the table simple, but we now give more details about the variables in the Methods text. (lines 255-262)

•Table 2: Include all of the coefficient estimates. Also, why 75%? I suggest including all, and bolding only those significant at 95% level. Also, which model do these estimates come from? Model 3? Are they consistent with model 6 or any of the others with random effects?

To our knowledge, there are no strict conventions for which central credible intervals to use. But we appreciate this suggestion and we now give all the parameter estimates and bold the ones with strong statistical support. We also now state more clearly for which model these results are, i.e. model variant 2 in the revised manuscript. The results between the model variant with and without random effects did correspond, but the variant completely without random effects was removed from the revised manuscript.

•Figure 3 (lines 666-669): Unclear why MCMC samples are being used to generated these estimates. Can't the estimates of co-infections be obtained directly using parametric expressions? Again, parametric expressions from the model would help.

We use the MCMC samples in order to include also the parameter uncertainty in the predictions. Our reasoning behind this figure is that simulated realizations of virus (co-) infections illustrate the performance of the model in terms of capturing the signal in the data. Parametric expressions often remain abstract for broader ecological audience, but we hope that our revised explanations of the modelling framework are helpful for all audience.

Reviewers' Comments:

Reviewer #1:

Remarks to the Author:

This is the second time that I have reviewed this manuscript by Sallinen et al.

Sallinen et al transplant 4 genotypes of *Plantago* hosts into 4 natural populations, observe whether they get infected by 5 virus species, and attribute patterns of infection and coinfection to variation in host genotype, variation space (i.e., the 4 different populations), host size, and herbivory. They find non-random associations of certain viruses in certain host genotypes, and they attribute these associations to differences in host genotype and local population context (i.e., space). A key result is that host genotype matters for the assembly of pathogen communities. As in my first review, I am excited about the experiment, its novel design, and its interesting results. Most of my previous concerns have been addressed. However, I still have two larger concerns, along with several specific questions below. Most of my concerns are about the analyses and their presentation.

Major

-1- Narrative: Attributing nonrandom coinfections to variation in space and host genotype.

One of the central results is that JSDM attributed the nonrandom patterns of coinfection to variation in host genotype and space. This is an interesting result. However, it has taken me two rounds of review, several close readings, and external readings about JSDM to understand it. This explanation was clearly described in the author's rebuttal to my first review (thank you!), and also stated in the discussion (line 450). However, it could be more clearly explained throughout the manuscript, especially in the introduction, methods, and results. I suggest explicitly introducing ideas in the introduction for how spatial variation and heterogeneity in host susceptibility could generate non-random co-occurrences, and more narrative built around the two statistical approaches, and how the results of the co-occur analysis (which identify the nonrandom patterns of coinfection) leads to JSDM for more refined hypothesis testing about these patterns (see specific points that confused me in text highlighted below).

-2- Description of analyses could still be clearer.

Although improved, I still found the description of the analyses confusing in places. I was not familiar with JSDM before reading this manuscript, but I suspect that many readers will be equally uninformed. Therefore, a premium ought to be placed on describing these models clearly and simply, with enough detail that quantitatively-trained ecologists can follow what you did and how you draw your conclusions. Details that I would like to see described more, even if briefly, include more information about the model competition (how to interpret explanatory vs. predictive power, and Tjur R²), what is meant by 'pooling' data from two time points, distributions responsible for p values in the co-occur analysis, nested vs. not nested random effects in JSDM, interpretation of low R² but 'nonsignificant' residuals (see specific comments below in Methods).

Line-by-line:

Abstract:

28: Would it be accurate to say something like "Using JSDM, we attribute the non-random co-occurrence patterns primarily to differences among host genotypes and local environmental context." A statement like this would clarify how the co-occur vs. JSDM results are related. (see Major 1)

Introduction

41: What do you mean by "the true diversity of infection?" This setup seems a little misleading, since you are targeting 5 viruses, not the entire microbiome.

55: "disease communities" is an odd phrase, as disease typically refers to host symptoms, not pathogen species. Pathogen communities?

67: often considerable variation?

70: What do you mean by higher trophic levels? Like herbivores and predators that consume hosts? I'm not following logic here.

83: What I'm missing from the introduction are more explicit ideas about how co-occurrence of pathogens could vary with spatial structure and host population structure. These alternative ideas – balancing the ideas of competition or facilitation among pathogens within hosts – would help prepare readers for your results. For example, heterogeneity in resistance among hosts could create non-random patterns of co-occurrence, as universally susceptible hosts could acquire diverse pathogen communities. Similarly, variation in infection risk across space could create non-random patterns of co-occurrence, as diverse pathogens could aggregate in hosts growing in 'hot-spots' that are generally favorable for multiple pathogens. Is this what you think is going on? If yes, developing these ideas seem essential for interpreting results. If no, then sorry, I must still be confused.

89: asymptomatic

92: I still think that it is confusing to call your spatial variable 'local population context'. I understand why you are resisting calling it 'space' (i.e., in order to identify these populations that have been historically studied and named consistently) but you could easily provide a table in the appendix that lists these historical names. I bring this up again, since other reviewers in round 1 also found this phrase confusing. I'm not 'requiring' this change, but noting that if multiple reviewers found it odd, then so might your readers.

101: This description of JSJM is still confusing. Perhaps part of the issue is that the paragraph focuses on what JSJMs enable from a statistical standpoint, without enough explanation of the biological puzzle that they allow you to address. Without reporting the co-occur results first, or developing ideas about how non-random co-occurrences could result from host heterogeneity and spatial heterogeneity, readers don't understand what needs to be "accounted for."

Methods

163: currently unknown

234: Pooled how? It's not clear what you mean. If both observations were used, then 'sampling time' ought to be included in the model, as the observations are not independent. If you collapsed the data onto one observation per plant, it's unclear how you treated a plant that was infected one time but not the other.

241: I still don't understand where these probabilities of co-occurrence come from – they must assume some underlying distribution. This explanation could be clearer. What is "the algorithm"?

Perhaps all this section needs is a reference to the appendix, but I also found the appendix unclear.

243: This transition to the JSJM framework, and its explanation, is still confusing. My diagnosis of what is missing (although I could be wrong) is some explanation that the co-occurrence patterns previously detected might result from a variety of mechanisms, including heterogeneity in space, variation in susceptibility among host genotypes, or interactions among viruses within hosts, and that JSJM allows you to evaluate support for these competing hypotheses.

267: Could be helpful to explicitly state what the comparison of model 1 vs. 2 tells you (i.e., importance of host genotype)

269: Is this specification of model variant 3 akin to a nested random effect of individual within genotype, or are you only allowing random differences among genotypes (not individuals?) It seems like if your goal is to evaluate whether host genotype adds any information about different residual co-occurrences, you would want the former. Typically, I'm more comfortable comparing nested models than models that differ in their structure. I still don't fully understand JSJM, so apologies if this comment doesn't make sense.

274: Are these model fit statistics penalized for model complexity, like AIC, or not? This seems important information for readers like me who are less familiar with JSJM and Tjur R². Of course, adding more terms to a model will increase its explanatory power; what is missing is some statement about how readers can interpret increases in Tjur R² with increasing model complexity (like qualitative

rules of thumb for interpreting delta AIC). Similarly, I also do not think that most readers will understand the difference between 'explanatory' and 'predictive' power – a brief description could be helpful.

Results

340: I would prefer to see this section preceded by a statement of the total variation explained by the best model (16%?)

I find it curious that the predictive power of the model is relatively low (16%, if I am interpreting that correctly), so presumably residual variation is 84%, yet "none of the residual correlations between virus species gained strong statistical support." Does this mean that the model isn't great at predicting individual virus occurrences, but that the model's prediction of each virus independently is consistent with the observed co-occurrence patterns? Sorry if this comment is obtuse, but I wonder if other readers with limited understanding of JSDM will be similarly confused.

Discussion

449: How many of these studies were conducting in the lab with controlled inoculations, versus in the field with presumably continuous and repeated exposure to multiple pathogens? Continuous exposure in the field seems likely to have contributed to the observation of weak interactions within hosts.

469: queues

Figures & Tables

735: Some definition of Tjur R2 and how to interpret it could be helpful, either as a footnote here or in the main text.

738: Why 90% instead of 95%? Since the Bayesian approach is arbitrary as you argue in your rebuttal, why not match the similarly arbitrary cutoff of 95% that is typically used in frequentist models? Similar comment for 90% cutoffs in the co-occur results, which seem to be derived from frequentist statistics (Figure 2).

772: Overlaying results onto the disease triangle seems forced, especially as you are putting host characteristics (effect of host individual) in the corner of the pathogen. It also highlights again, how it could be clearer to talk about 'host population' as 'environment' or 'space', as 'host population' clearly implies something about the host.

Appendix

36: "without reference to a statistic"... confused. There must be some underlying distribution that you are comparing to in order to get p values, right? Sorry if I am missing something here.

39: Why don't right columns sum to 1? Something doesn't 'add up' here; sorry if I am missing something obvious.

55: ran

I hope that you find these reviews helpful.

Signed,

Alex Strauss

Reviewer #2:

Remarks to the Author:

The authors have done a thorough job of responding to the reviewer comments, and have strengthened the manuscript considerably. There are a few minor points of clarification that would be helpful and could be easily addressed.

Line 178-181. The added sentence helps, but the terminology is still challenging. We have 'target populations', 'mother plants', and 'transplant populations'. I am not sure from the terms alone which is which. Please rephrase to make the relationships super clear. We want to be able to easily grasp the structure of this valuable work.

Lines 235-237. I find it hard to parse the clause 'To understand whether the viruses occur more or less than would be expected based on their frequencies....'

Instead, what about 'To understand whether the probability of virus co-occurrence differs from that which would be expected based solely on virus prevalence'? This may not be the best revision, but consider adjusting this clause in some way. I have similar challenges with lines 323-325.

Lines 255-262. Since 'local plant population context' is such a key concept here and one that tripped up all the reviewers in its original wording, how about adding one more sentence at the end of the para. to remind us explicitly what effects might be wrapped up in it?

Paragraph starting line 264. Can you guide the reader by explicitly stating what insight will be gained by comparing models 2 and 3?

Lines 466—468. '...we kept the plants in their pots which is likely to generate different nutrient and soil microbial communities than those experienced by the wild plants. However, this approach allowed us to control for this level of variation in our data.'

I think it's really important to consider that additional differences (e.g., soil moisture, rooting depth, root allocation, root structure...) come into play when comparing pot-grown to field-grown plants. Can you perhaps acknowledge this more simply but also more broadly by saying something like "we kept the plants in their pots, which meant these plants experienced different rooting environments than the wild plants but allowed us to standardize some factors (e.g., soil medium)." I don't think that using the same potting medium completely controls for variation in soil microbe communities since those could be seeded in the field, so I suggest this simpler phrasing.

Line 470 – 472. 'Nonetheless, the sentinel plants were hosts to all five focal viruses suggesting that transmission to them was similar as to the naturally growing host plants in these populations.'

I strongly disagree with the last part of this sentence; from the data presented, we don't know how similar transmission was to that in naturally growing plants. Instead, how about something more straightforward, such as 'Nonetheless, transmission of all five focal viruses to the sentinel plants did occur.'

Table 1. To aid reader understanding, consider rewriting the table header as "Joint species distribution model variants, their explanatory power (measured by Tjur R2 coefficient of determination), and predictive power (similar coefficient of determination derived from cross-validation analysis, see Methods and Supplementary)."

Please replace the term 'wild population' with the terminology as used in the body of the manuscript. I think you mean 'local population context'.

A few small typos:

Line 101 – not clear what "they" refers to. Maybe "these patterns"?

Lines 111-120, add question marks at end of the questions

Line 144: 'called by' better as 'referred to by'

Line 163: missing final n on unknown

Line 171 'After one month, the roots */that had/* grown from....'

Line 460 'short-lasing' should be 'short-lasting'

Line 466. 'post' should be 'pots'

Line 469, 'ques' should be 'cues'

Best wishes, Carolyn Malmstrom

Reviewer #3:

Remarks to the Author:

See attached.

Review of “Intraspecific host variation plays a key role in virus community assembly”

The authors conducted a thorough revision of their original manuscript in response to the suggestions posed by the three referees. They have added a lot of details that were originally missing from their initial submission. A few remaining comments are given in detail below. As I mention, I still find it imperative that the authors give a more formal description of the models they are fitting. It wouldn't take much space to describe the different components of L_{ij} (as identified currently in the supplement) and would make it much more clear to the reader the difference between the models.

- From my previous review (and now lines 194-196): Is there evidence to support that using the averages in place of the missing values has no impact on the results?
- Line 251: You mention that the latent factors can account for the spatio-temporal structure of the data. I thought you aggregated across time?
- How does the latent variable approach account for residual variance *after* accounting for the fixed effects? Isn't the model being fitting all at once or is there some form of two stage process here?
- Line 256: Why define \mathbf{Y} and \mathbf{X} if you aren't going to write out the model? I highly recommend that you reconsider writing out the model components like you have in the supplement (with $L_{ij} = L_{ij}^F + L_{ij}^R + \epsilon_{ij}$) as a way to depict the differences between the 3 models you are considering. Then, you can define $i = 1, \dots, ?$ and $j = 1, \dots, ?$ to remind the reader the number of plants and viruses, and the different specifications of both the fixed and random effects. As it reads, I am still unable to decipher what the three models are that are being compared. This would help to define the number of random effects in the models and answer my question below about how much these random effects vary (presumably a lot?)
- What are the estimates of the variance parameters of the latent variables? These are likely assigned some form of inverse-gamma prior and have posterior distributions. These should be summarized in the text (if not in the table).
- How sensitive are your results to the default priors of the R packages you used?
- In response to the other reviewers, it seems reasonable to also compare the models based on DIC.
- Since Tjur R^2 isn't extremely common, the authors might consider describing in more detail the significance of the values in Table 1. If someone interpreted these like a typical R^2 , values of 0.16 and below would suggest the models are not very good.

Revision summary

We thank all the reviewers for their helpful and constructive comments which have significantly improved our manuscript. In this revision, we have especially focused on improving our communication of the differences, biological meaning, and mathematical background of the co-occurrence analysis and JSJM. Moreover, we did additional analysis with different prior parameters to check how this influenced our results, and these results are now also reported in the manuscript. Finally, we have further clarified the terminology of our manuscript. We have specifically:

1. We have revised our description of JSJM modelling throughout the manuscript to better link our biological research questions and the statistical methods. We have added detailed explanation of the mathematical background to the statistical methods.
2. We have clarified the section of model comparison and now include more detailed information about the calculation and interpretation of $T_{jur}R^2$. We now also complement the $T_{jur}R^2$ comparison with comparison of Widely Applicable Information Criterion (WAIC).
3. We have fit new JSJM models with less conservative priors to assess the robustness of our results to the choice of priors. We report that the main findings are robust for the choice of priors, but our new results show that with more sensitive priors for the latent variables, we detect one virus-virus association with strong statistical support. We conclude that although there is potential for deriving residual associations in these virus communities, the resolution and size of our data set does not allow for making any other conclusions. Hence, we report as our main findings the results using default priors which we consider to be a more conservative analysis of our relatively small dataset, but we have updated the Methods and Results to also report results of the model using the less conservative priors.
4. We have revised the text overall for better consistency in terminology and further clarified our communication of what we think "local population context" includes, i.e. it includes the characteristics of the host plant population such as the distinct histories of these populations.

Reviewer #1 (Remarks to the Author):

This is the second time that I have reviewed this manuscript by Sallinen et al.

Sallinen et al transplant 4 genotypes of *Plantago* hosts into 4 natural populations, observe whether they get infected by 5 virus species, and attribute patterns of

infection and coinfection to variation in host genotype, variation space (i.e., the 4 different populations), host size, and herbivory. They find non-random associations of certain viruses in certain host genotypes, and they attribute these associations to differences in host genotype and local population context (i.e., space). A key result is that host genotype matters for the assembly of pathogen communities. As in my first review, I am excited about the experiment, its novel design, and its interesting results. Most of my previous concerns have been addressed. However, I still have two larger concerns, along with several specific questions below. Most of my concerns are about the analyses and their presentation.

We thank the reviewer for these kind words. We have revised our manuscript to address these concerns, and respond to the specific requests below.

Major

-1- Narrative: Attributing nonrandom coinfections to variation in space and host genotype.

One of the central results is that JSDM attributed the nonrandom patterns of coinfection to variation in host genotype and space. This is an interesting result. However, it has taken me two rounds of review, several close readings, and external readings about JSDM to understand it. This explanation was clearly described in the author's rebuttal to my first review (thank you!), and also stated in the discussion (line 450). However, it could be more clearly explained throughout the manuscript, especially in the introduction, methods, and results. I suggest explicitly introducing ideas in the introduction for how spatial variation and heterogeneity in host susceptibility could generate non-random co-occurrences, and more narrative built around the two statistical approaches, and how the results of the co-occur analysis (which identify the nonrandom patterns of coinfection) leads to JSDM for more refined hypothesis testing about these patterns (see specific points that confused me in text highlighted below).

We thank the reviewer for these comments. We have elaborated on the JSDM in the introduction and methods according to the specific points listed below.

-2- Description of analyses could still be clearer.

Although improved, I still found the description of the analyses confusing in places. I was not familiar with JSDM before reading this manuscript, but I suspect that many readers will be equally uninformed. Therefore, a premium ought to be placed on describing these models clearly and simply, with enough detail that quantitatively-trained ecologists can follow what you did and how you draw your conclusions.

Details that I would like to see described more, even if briefly, include more information about the model competition (how to interpret explanatory vs. predictive power, and Tjur R²), what is meant by ‘pooling’ data from two time points, distributions responsible for p values in the co-occur analysis, nested vs. not nested random effects in JSDM, interpretation of low R² but ‘nonsignificant’ residuals (see specific comments below in Methods).

We thank the reviewer for these comments. We now elaborate on how we implemented the model comparison (starting from line 533), as well as on how we pooled the data (lines 438-442). In addition, we changed the expression “power” to “performance”, hoping it is more informative and intuitive. We also went through the specific points raised by the reviewer (below) and hope that we now make these definitions and connections clearer.

Line-by-line:

Abstract:

28: Would it be accurate to say something like “Using JSDM, we attribute the non-random co-occurrence patterns primarily to differences among host genotypes and local environmental context.” A statement like this would clarify how the co-occur vs. JSDM results are related. (see Major 1)

Thank you for this suggestion, we have edited the abstract accordingly: *“Using joint species distribution modelling, we attribute the non-random co-occurrence patterns primarily to differences among host genotypes and local population context.”*

Introduction

41: What do you mean by “the true diversity of infection?” This setup seems a little misleading, since you are targeting 5 viruses, not the entire microbiome.

We thank the reviewer for this comment. We have toned down this sentence and removed the “true” (line 39). We now say: *“Consequently, accounting for the diversity of infection is necessary to understand and predict disease dynamics and costs of infection for the host.”*

55: “disease communities” is an odd phrase, as disease typically refers to host symptoms, not pathogen species. Pathogen communities?

We thank the reviewer for noticing this inaccuracy and have changed the word “disease” to “parasite”. (line 52)

67: often considerable variation?

We thank the reviewer for this suggestion and have added “often” to the sentence. (line 65)

70: What do you mean by higher trophic levels? Like herbivores and predators that consume hosts? I’m not following logic here.

We mean the organisms that exploit the host resources for their own growth and reproduction. We have modified the sentence accordingly: *“However, the importance of intraspecific host resistance variation for community assembly and diversity of species that exploit the host is only beginning to gain attention.”* (line 68)

83: What I’m missing from the introduction are more explicit ideas about how co-occurrence of pathogens could vary with spatial structure and host population structure. These alternative ideas – balancing the ideas of competition or facilitation among pathogens within hosts – would help prepare readers for your results. For example, heterogeneity in resistance among hosts could create non-random patterns of co-occurrence, as universally susceptible hosts could acquire diverse pathogen communities. Similarly, variation in infection risk across space could create non-random patterns of co-occurrence, as diverse pathogens could aggregate in hosts growing in ‘hot-spots’ that are generally favorable for multiple pathogens. Is this what you think is going on? If yes, developing these ideas seem essential for interpreting results. If no, then sorry, I must still be confused.

We thank the reviewer for this helpful comment. To further elaborate on how co-occurrence patterns may vary in space, we have added a sentence and now say (lines 79-82): *“Variation in host resistance may be spatially structured with pronounced differences in resistance observed among host populations (Jousimo et al. 2014) and regions (Burdon et al. 1999). Such spatially structured resistance variation may also drive spatially structured co-occurrence patterns of pathogens exploiting the same host.”* Our aim is indeed to tease apart the effect individual host genotypes from other drivers of virus occurrence. Our study design where we replicated the host genotypes in each of the four study populations and mixed the locations of our individual sentinel plants every few days allowed us to estimate the effect of host genotype on pathogen co-occurrence patterns.

89: asymptomatic

Thank you for noticing this typo, we have corrected it. (line 90)

92: I still think that it is confusing to call your spatial variable ‘local population context’. I understand why you are resisting calling it ‘space’ (i.e., in order to

identify these populations that have been historically studied and named consistently) but you could easily provide a table in the appendix that lists these historical names. I bring this up again, since other reviewers in round 1 also found this phrase confusing. I'm not 'requiring' this change, but noting that if multiple reviewers found it odd, then so might your readers.

We thank the reviewer for this comment. However, we decided to stick with our choice and use "local population context". We feel that including the term "population" is essential because we want the term to deliver the idea that the features of the host population, such as host population genetic structure, demography, and the historical pathogen pools these host populations have encountered, together with environmental factors (e.g. variation in plant and vector communities), are included in the underlying processes that may affect the virus communities in these populations. We feel that "local population context" captures the myriad of these factors better than "space" or "environment". The *P. lanceolata* populations are indeed distinct, fragmented, small, and clearly delimited. We have added the host population features to the definition of this concept in the Introduction (line 94): "...-within the local population context, which may include environmental variation, the local disease pool, host population structure and history, as well as local vector communities.-..." and elaborated on this in the Discussion (lines 246-248): "*The local population context further includes any differences in population dynamics and trajectories, such as historical pathogen pressure, which may vary among these populations (Ojanen et al. 2013)*".

101: This description of JSJM is still confusing. Perhaps part of the issue is that the paragraph focuses on what JSJMs enable from a statistical standpoint, without enough explanation of the biological puzzle that they allow you to address. Without reporting the co-occur results first, or developing ideas about how non-random co-occurrences could result from host heterogeneity and spatial heterogeneity, readers don't understand what needs to be "accounted for."

We have now considerably revised this paragraph, and hope that it now more clearly describes how the methods can help understand the biological questions of our study (starting from line 98).

Methods

163: currently unknown

Thank you for pointing out this mistake, we have corrected it. (line 368)

234: Pooled how? It's not clear what you mean. If both observations were used, then 'sampling time' ought to be included in the model, as the observations are not

independent. If you collapsed the data onto one observation per plant, it's unclear how you treated a plant that was infected one time but not the other.

We thank the reviewer for pointing out this unclarity. We mean that the observations were collapsed. We thus had one observation (virus community) per plant. We have clarified this in the manuscript by stating (lines 438-442): *“For all the statistical analysis, we pooled the detected occurrences of the five focal viruses over the two timepoints of sampling by collapsing the occurrence data so that each plant had one observed virus community. Only when a plant had not been infected by a certain virus in either of the timepoints accounted as an absence of the virus while infection in one or both timepoints was accounted for as virus presence.”*

241: I still don't understand where these probabilities of co-occurrence come from – they must assume some underlying distribution. This explanation could be clearer. What is “the algorithm”? Perhaps all this section needs is a reference to the appendix, but I also found the appendix unclear.

Following this comment, as well as the comment by Reviewer 2 on this same phrase, we have modified it to the following (lines 442-444): *“To understand whether the co-occurrences of viruses differs from expected co-occurrences calculated solely from the prevalences of these viruses,-...”*. We also expanded the last sentence of the paragraph to clarify the used method (lines 448-453): *“By comparing the expected and observed co-occurrences the applied algorithm gives the probabilities of co-occurrence greater than or less than what is observed in the data analytically, without relying on randomisations or test statistics, under the condition that the probability of occurrence for a species at each sentinel plant is equal to its observed frequency among all the sentinel plants, i.e. in this case the prevalence of the virus (Veech 2013)”*. We also cite the original publication for the method (Veech 2013) and its implementation (Griffith *et al.* 2016).

243: This transition to the JSMD framework, and it's explanation, is still confusing. My diagnosis of what is missing (although I could be wrong) is some explanation that the co-occurrence patterns previously detected might result from a variety of mechanisms, including heterogeneity in space, variation in susceptibility among host genotypes, or interactions among viruses within hosts, and that JSMD allows you to evaluate support for these competing hypotheses.

We have revised the Introduction to our motivation for selecting this modeling framework: *“We first test whether the viruses occur in the same host more often than would be expected based on their frequencies alone. In other words, we test whether virus co-occurrence patterns differ from expectations of a random distribution. We then employ a joint species distribution modelling (JSMD) framework (Warton *et al.* 2015), that allows us to tease apart the effect of local population context (consisting of unmeasured environmental variation as well as host population structure and history) on virus (co-)occurrences from host plant characteristics and host genotype. We can account for the*

shared environmental responses of the target species, which makes the model a robust method also for sparse data (Ovaskainen & Soininen 2011). Using this approach, we are also able to capture signals of possible biotic community assembly processes from virus-to-virus association matrices after controlling for shared environmental responses of the viruses.” We have also strengthened the link between our study questions and methods by starting our paragraph in the Methods on line 454 by referring to our study questions: “To address our study questions on how the host genotype and characteristics, as well as local population context affect the (co-)occurrence patterns of the viruses, as well as the possible signals of biotic interactions between the viruses affecting their distribution, we applied a joint species distribution modelling (JSDM) framework...”

267: Could be helpful to explicitly state what the comparison of model 1 vs. 2 tells you (i.e., importance of host genotype)

We have added this detail starting from line 533: *“The model comparison approach allows us to examine the relevance of sentinel plant genotype as a predictor of virus community composition (comparison of model variants 1 and 2), as well as to see whether the residual co-occurrences between the viruses differ between the sentinel plant genotypes (variant 3).”*

269: Is this specification of model variant 3 akin to a nested random effect of individual within genotype, or are you only allowing random differences among genotypes (not individuals?) It seems like if your goal is to evaluate whether host genotype adds any information about different residual co-occurrences, you would want the former. Typically, I’m more comfortable comparing nested models than models that differ in their structure. I still don’t fully understand JSDM, so apologies if this comment doesn’t make sense.

The reasoning behind our approach is to see which representation of the virus community is the best one in terms of the performance of the model. We are not only interested in which covariates explain the observed virus occurrences but also in which way do they explain them the best: are the effects of the genotypes uniform, or do they differ in different host populations. The difference between variants 2 and 3 resembles the inclusion of interaction terms in the model, but with our approach we can do this by specifically looking at the residual correlations.

274: Are these model fit statistics penalized for model complexity, like AIC, or not? This seems important information for readers like me who are less familiar with JSDM and Tjur R2. Of course, adding more terms to a model will increase its explanatory power; what is missing is some statement about how readers can interpret increases in Tjur R2 with increasing model complexity (like qualitative rules of thumb for interpreting delta AIC). Similarly, I also do not think that most readers will understand the difference between ‘explanatory’ and ‘predictive’ power – a brief description could be helpful.

We do not see a problem in the comparison of the models with the coefficient of determination only, because we are looking at both explanatory and predictive powers, and instead of an overfit, we actually see better predictive power for the model variant 2. This would be an issue if we would only be looking at explanatory power, but if we would be overfitting with an excessive number of variables, this would be seen in a dramatic decrease in the predictive power of the models. We now also elaborate on what predictive and explanatory performance mean (starting from line 537): *“We compared the model variants in terms of their performance, where the first tells us how well the model predicts the data used to fit it, whereas the latter illustrates how well the model predicts independent data which has not been used for model fitting.”* We now also complemented our results by calculating the Widely Applicable Information Criterion (WAIC), which is the recommended measure for HMSC (Ovaskainen & Abrego 2020). The results are now reported in Table 1. and we now describe the WAIC calculation in the methods (lines 553-555) as well as results (line 159 and Table 1). In summary, model variant 2 gained the lowest WAIC value, which further supports its selection as our best model variant, although the differences between models 2 and 3 were very minor in terms of the WAIC.

We now elaborate also on the interpretation of the Tjur R^2 , which is slightly different from traditional R^2 measures, on lines 540-553: *“We calculated the Tjur R^2 coefficient of determination, a statistic that has been recommended to be used as a standard measure of explanatory power for binary outcomes (Tjur 2009). The coefficient is obtained by calculating the mean of the predicted probabilities of presences and absences, and then taking the difference between those two means. Hence, a high coefficient value implies high predicted probabilities for presences and low probabilities for absences. When interpreting it, it is good to note that with sparse data, the probabilities of presence tend to be low in the first place, and thus the Tjur R^2 coefficient can remain rather low as well. Nevertheless, if the model is completely uninformative and predicts a 50% probability for both presence and absence, the coefficient value will be zero, thus revealing a poor model fit. For examining explanatory power, we fit the model to the full data set and base our comparison on predictions made for the same data. To examine the predictive power of the model, we conducted a 10-fold cross-validation and compared the model variants based on the same Tjur R^2 coefficient as with explanatory power, but calculated from the predictions made to new, unknown host plants.”*

Results

340: I would prefer to see this section preceded by a statement of the total variation explained by the best model (16%?). I find it curious that the predictive power of the model is relatively low (16%, if I am interpreting that correctly), so presumably residual variation is 84%, yet “none of the residual correlations between virus species gained strong statistical support.” Does this mean that the model isn’t great at predicting individual virus occurrences, but that the model’s prediction of each virus independently is consistent with the observed co-occurrence patterns? Sorry if this comment is obtuse, but I wonder if other readers with limited understanding

of JSDM will be similarly confused.

Ecological data is inherently noisy, and thus it is quite expected to have unexplained variance in the data. What we see with our modelling approach, is that we can explain some of the variance with our fixed effects, and of the remaining variance, we can explain some with our random effects, but after this, there remains unexplained variation still. We now explain in more detail the Tjur R^2 coefficient of determination (line 540), as outlined in our previous answer, which we hope to clarify this issue further. We chose to use this coefficient because of its easy interpretation and we hope this comes across to the reader now also. Based on our results, our models predict moderate occurrence probabilities across the whole data, and this results in the low average Tjur R^2 value. Please note also, that we do not rely only on this, but we also display clearly how well the model simulates the data (Figure 3).

Discussion

449: How many of these studies were conducting in the lab with controlled inoculations, versus in the field with presumably continuous and repeated exposure to multiple pathogens? Continuous exposure in the field seems likely to have contributed to the observation of weak interactions within hosts.

These references include review articles that describe both controlled inoculation studies and field experiments. We also provide examples of field experiments where strong evidence of interactions has been detected: Telfer *et al.* 2010 Science; Laine 2011 Evol. Appl. ; Halliday *et al.* 2020 Nature Ecol Evol.

469: queues

Thank you for noticing there is a typo. We have corrected the word to “cues”. (line 311)

Figures & Tables

735: Some definition of Tjur R^2 and how to interpret it could be helpful, either as a footnote here or in the main text.

We added a sentence to the main text, starting from line 540: *“We calculated the Tjur R^2 coefficient of determination, a statistic that has been recommended to be used as a standard measure of explanatory power for binary outcomes (Tjur 2009). The coefficient is obtained by calculating the mean of the predicted probabilities of presences and absences, and then taking the difference between those two means. Hence, a high coefficient value implies high predicted probabilities for presences and low probabilities for absences. When interpreting it, it is good to note that with sparse data, the probabilities of*

presence tend to be low in the first place, and thus the Tjur R^2 coefficient can remain rather low as well. Nevertheless, if the model is completely uninformative and predicts a 50% probability for both presence and absence, the coefficient value will be zero, and thus revealing a poor model fit.”

738: Why 90% instead of 95%? Since the Bayesian approach is arbitrary as you argue in your rebuttal, why not match the similarly arbitrary cutoff of 95% that is typically used in frequentist models? Similar comment for 90% cutoffs in the co-occur results, which seem to be derived from frequentist statistics (Figure 2).

As we noted before, the choice of the limit is indeed quite arbitrary, and our choice of 90% is quite conservative compared to for example a 75% cutoff reported in e.g. Dallas *et al.* 2019. Our results are not particularly sensitive to the limit chosen, nor do we base our results and conclusions heavily on these ‘significances’. With our results we show that there are patterns and correlations which gain strong statistical support, and we look into these more closely with the variance partitioning, but we do not rely on any individual ‘significances’.

772: Overlaying results onto the disease triangle seems forced, especially as you are putting host characteristics (effect of host individual) in the corner of the pathogen. It also highlights again, how it could be clearer to talk about ‘host population’ as ‘environment’ or ‘space’, as ‘host population’ clearly implies something about the host.

We thank you for this comment, which highlights that our figure was not clear enough. The ‘effect of host individual’ was misleading because it actually means the random effects at the host individual level, i.e. residual correlations between viruses, which can be considered as possible signals of pathogen-pathogen interactions, as explained in the main text. This is why this effect is in the corner of the pathogen. The known effects of the host plant are in the ‘host’ corner. We corrected the wording in this figure and we hope it is now more clear and illustrative.

Appendix

36: “without reference to a statistic”... confused. There must be some underlying distribution that you are comparing to in order to get p values, right? Sorry if I am missing something here.

The algorithm we are using is based on combinatorics. No randomisations or test statistics are calculated. As it says in the original research article of the method (Veech 2013), the model allows one to analytically (i.e. without randomization or simulation) obtain the probability that two selected species co-occur at a frequency either less than or greater than the observed frequency of co-occurrence. We refer to this publication in the main text, and we also now include a more elaborate explanation on the method, starting from line 448.

39: Why don't right columns sum to 1? Something doesn't 'add up' here; sorry if I am missing something obvious.

As described in the original paper by Veech (2013), the significance values are gained by summing the probability of less (or more, respectively) co-occurrences than would be expected based on the frequency of the focal species and the probability that by chance the observed co-occurrence is exactly equal to what would be expected based on the frequencies.

55: ran

Thank you for noticing the typo. We have corrected it.

Reviewer #2 (Remarks to the Author):

The authors have done a thorough job of responding to the reviewer comments, and have strengthened the manuscript considerably. There are a few minor points of clarification that would be helpful and could be easily addressed.

We thank the reviewer for these kind words. We have clarified the points noted and our responses appear point-by-point below.

Line 178-181. The added sentence helps, but the terminology is still challenging. We have 'target populations', 'mother plants', and 'transplant populations'. I am not sure from the terms alone which is which. Please rephrase to make the relationships super clear. We want to be able to easily grasp the structure of this valuable work.

We thank the reviewer for this helpful comment. We have simplified the text by deleting these unnecessary and confusing words: "target" and "transplant" and "mother". We now consistently use the following terminology: 1) Plants that were used to clone the sentinel plants are referred to as maternal plants, 2) The plants used in the field experiment are referred to as sentinel plants, 3) The sites of the transplant experiment are referred to as *P. lanceolata*- or plant populations or 'local population context' depending on the context.

Lines 235-237. I find it hard to parse the clause 'To understand whether the viruses occur more or less than would be expected based on their frequencies....'. Instead, what about 'To understand whether the probability of virus co-occurrence differs from that which would be expected based solely on virus prevalence'? This may not

be the best revision, but consider adjusting this clause in some way. I have similar challenges with lines 323-325.

We thank the reviewer for this helpful comment. To improve the understandability, we have now modified the text as follows: *"To understand whether the co-occurrence of viruses differs from expected co-occurrences calculated solely from the prevalences of these viruses,-..."* (line 442)

Lines 255-262. Since 'local plant population context' is such a key concept here and one that tripped up all the reviewers in its original wording, how about adding one more sentence at the end of the para. to remind us explicitly what effects might be wrapped up in it?

We have added a description to the methods of what this variable represents on lines 499-502 and say: *"As explanatory variables (denoted by matrix X in (Ovaskainen et al. 2017)) we used the local plant population context (categorical variable with four classes), which is a proxy for the plant population level effects such as variation in abiotic conditions, vector communities, and disease pool;-..."* We also describe this in the Introduction on lines 93-95 and 102-104.

As the sentence on line 255-256 has caused confusion also to the other Reviewers, we have removed it and moved the notion that latent variables are random effects to the previous sentence.

Paragraph starting line 264. Can you guide the reader by explicitly stating what insight will be gained by comparing models 2 and 3?

We have revised thoroughly this part of the methods, and we now state (starting from line 533) that: *"The model comparison approach enables us to look into the relevance of host genotype as a predictor of virus community composition (comparison of model variants 1 and 2), as well as to see whether the residual co-occurrences between the viruses differ between the host plant genotypes (variant 3)."*

Lines 466—468. '....we kept the plants in their pots which is likely to generate different nutrient and soil microbial communities than those experienced by the wild plants. However, this approach allowed us to control for this level of variation in our data.'

I think it's really important to consider that additional differences (e.g., soil moisture, rooting depth, root allocation, root structure...) come into play when comparing pot-grown to field-grown plants. Can you perhaps acknowledge this more simply but also more broadly by saying something like "we kept the plants in their pots, which meant these plants experienced different rooting environments than the wild plants but allowed us to standardize some factors (e.g., soil medium)." I don't think that using the same potting medium completely controls for variation in

soil microbe communities since those could be seeded in the field, so I suggest this simpler phrasing.

Thank you for suggesting this more general and simpler sentence, we have edited the text accordingly. (lines 307-309)

Line 470 – 472. ‘Nonetheless, the sentinel plants were hosts to all five focal viruses suggesting that transmission to them was similar as to the naturally growing host plants in these populations.’

I strongly disagree with the last part of this sentence; from the data presented, we don’t know how similar transmission was to that in naturally growing plants. Instead, how about something more straightforward, such as ‘Nonetheless, transmission of all five focal viruses to the sentinel plants did occur.’

Thank you for this very helpful suggestion, we have modified the sentence to the suggested one. (lines 311-312)

Table 1. To aid reader understanding, consider rewriting the table header as “Joint species distribution model variants, their explanatory power (measured by Tjur R² coefficient of determination), and predictive power (similar coefficient of determination derived from cross-validation analysis, see Methods and Supplementary).” Please replace the term ‘wild population’ with the terminology as used in the body of the manuscript. I think you mean ‘local population context’.

We unified the terminology as suggested, thank you for noticing this. We also thank you for the suggestion for the header, and we now changed it accordingly to as follows: “*Joint species distribution model variants and their explanatory performances and predictive performances (based on cross-validation), measured by the Tjur R² coefficient of determination (see Methods and Supplementary Information)*”.

A few small typos:

Line 101 – not clear what “they” refers to. Maybe “these patterns”?

Lines 111-120, add question marks at end of the questions

Line 144: ‘called by’ better as ‘referred to by’

Line 163: missing final n on unknown

Line 171 ‘After one month, the roots */that had/* grown from....’

Line 460 ‘short-lasing’ should be ‘short-lasting’

Line 466. ‘post’ should be ‘pots’

Line 469, ‘ques’ should be ‘cues’

We thank for pointing out these typos and have corrected them. For the typo on line 375 we have written “*the new plants shooting from the cut roots*”.

Reviewer #3 (Remarks to the Author):

The authors conducted a thorough revision of their original manuscript in response to the suggestions posed by the three referees. They have added a lot of details that were originally missing from their initial submission. A few remaining comments are given in detail below. As I mention, I still find it imperative that the authors give a more formal description of the models they are fitting. It wouldn't take much space to describe the different components of L_{ij} (as identified currently in the supplement) and would make it much more clear to the reader the difference between the models.

•From my previous review (and now lines 194-196): Is there evidence to support that using the averages in place of the missing values has no impact on the results?

Our modelling method does not allow for missing values in the explanatory variables, but because the responses of species are estimated so that information is borrowed across species, and our prior is normal, we consider this choice as conservative and sound. There were in total 13 plants for which we used the overall means. These plants are distributed across all populations and genotypes.

•Line 251: You mention that the latent factors can account for the spatio-temporal structure of the data. I thought you aggregated across time?

We thank for pointing out this confusing sentence. Here, we were describing the general possibilities of the framework used, but indeed it may be confusing for the reader, and hence we have deleted this part.

•How does the latent variable approach account for residual variance after accounting for the fixed effects? Isn't the model being fitting all at once or is there some form of two stage process here?

HMSC is a multivariate hierarchical generalized linear mixed model fitted with Bayesian interface. During each MCMC iteration, all parameters are updated. Our expression underlies that we account for fixed (environmental) effects, and only the residual is used for the latent part.

•Line 256: Why define Y and X if you aren't going to write out the model? I highly recommend that you reconsider writing out the model components like you have in the supplement (with $L_{ij} = L_{fij} + L_{rij} + ij$) as a way to depict the differences

between the 3 models you are considering. Then, you can define $i= 1, \dots, ?$ and $j= 1, \dots, ?$ to remind the reader the number of plants and viruses, and the different specifications of both the fixed and random effects. As it reads, I am still unable to decipher what the three models are that are being compared. This would help to define the number of random effects in the models and answer my question below about how much these random effects vary (presumably a lot?)

The modelling method we use is quite complex to write out in full detail, due to its hierarchical structure. We defined variables so that they correspond to the notation in the primary publications describing the model structure in detail, both in terms of statistics and community ecological interpretations. We now include a more formal description of the model in the methods (lines 469-496), not just the appendix. We hope that the current level of specificity is enough, as we make clear references to both the original article introducing the method as well as to the recently published book (Ovaskainen & Abrego 2020) describing the method in detail. Throughout we use the same notation, so that all the equations and explanations in the book are applicable. We feel it is beyond the scope of the methods section of this manuscript to describe the full structure of the model, since although it is a novel method, it is well-documented and its function well-established.

•What are the estimates of the variance parameters of the latent variables? These are likely assigned some form of inverse-gamma prior and have posterior distributions. These should be summarized in the text (if not in the table).

Thank you for this helpful suggestion. We have revised the text to clarify (starting from line 479) that the latent variables consist of site and species loading, which in our data are sentinel plants and virus specie, both of which are estimated. The variance-covariance matrix shows both the covariances between species as well as within-species variances. The prior distributions for the latent variables are described in detail in Ovaskainen & Abrego (2020) and Ovaskainen *et al.* (2017), and we clearly state in the text that we use the defaults of the framework.

•How sensitive are your results to the default priors of the R packages you used?

In general, the HMSC models are robust regarding the choices of priors (see Ovaskainen *et al.* 2020), but the parameters influencing the shrinkage of the latent variables are the most sensitive part of the model. The default is a strongly shrinking prior and truncating the number of estimated factors to the number of species in the data set, hence in our case 5. We thank the reviewer for pointing this out. We decided to complement our current modelling pipeline by showing the influence of the choice of the prior for the latent variables. Our new results show that with more sensitive priors for the latent variables, we detect one virus-virus association with strong statistical support. Nevertheless, we do not detect any other interactions, and thus we conclude that although there is potential for deriving residual associations in these virus communities, the resolution and size of our data set does not allow for making any other conclusions. Hence, we report as our main findings the results using default priors which we consider to be a more conservative analysis of our relatively small dataset, but we have updated the Methods (lines 522-531)

and Results (lines 206-210) to also report results of the model using the less conservative priors.

•In response to the other reviewers, it seems reasonable to also compare the models based on DIC.

DIC does not perfectly fit the modelling method we apply (for a thorough explanation, see Chapter 9.3 in Ovaskainen *et al.* 2020). Instead, we complemented our results by calculating the Widely Applicable Information Criterion (WAIC), which is the recommended measure for HMSC. We now include WAIC calculation in the methods (lines 553-555) as well as results (line 159 and Table 1). In summary, model variant 2 gained the lowest WAIC value, which further supports its selection as our best model variant, although the differences between models 2 and 3 were very minor in terms of the WAIC.

•Since Tjur R² isn't extremely common, the authors might consider describing in more detail the significance of the values in Table 1. If someone interpreted these like a typical R², values of 0.16 and below would suggest the models are not very good.

We thank the reviewer for this helpful suggestion. We now include a more detailed description of the interpretation of our measure of choice. Starting from line 539, we state: “We calculated the Tjur R² coefficient of determination, a statistic that has been recommended to be used as a standard measure of explanatory power for binary outcomes (Tjur 2009). The coefficient is obtained simply by calculating the mean of the predicted probabilities of presences and absences, and then taking the difference between those two means. Hence, a high coefficient value implies high predicted probabilities for presences and low probabilities for absences. When interpreting it, it is good to note that with sparse data, the probabilities of presence tend to be low in the first place, and thus the Tjur R² coefficient can remain rather low as well. Nevertheless, if the model is completely uninformative and predicts a 50% probability for both presence and absence, the coefficient value will be zero, and thus revealing a poor model fit.”

References

- Burdon, A.J.J., Thrall, P.H. & Brown, A.H.D. (1999). Resistance and Virulence Structure in Two Linum marginale-Melampsora lini Host-Pathogen Metapopulations with Different Mating Systems. *Evolution (N. Y.)*, 53, 704–716.
- Dallas, T.A., Laine, A., Ovaskainen, O. & Dallas, T.A. (2019). Detecting parasite associations within multi-species host and parasite communities. *Proc. R. Soc. B Biol. Sci.*, 286, 20191109.
- Griffith, D.M., Veech, J.A. & Marsh, C.J. (2016). cooccur : Probabilistic Species Co-Occurrence Analysis in R. *J. Stat. Softw.*, 69, 1–17.

- Halliday, F.W., Penczykowski, R.M., Barrès, B., Eck, J.L., Numminen, E. & Laine, A.-L. (2020). Facilitative priority effects drive parasite assembly under coinfection. *Nat. Ecol. Evol.*, doi.org/10.
- Jousimo, J., Tack, A.J.M., Ovaskainen, O., Mononen, T., Susi, H., Tollenaere, C., *et al.* (2014). Disease ecology. Ecological and evolutionary effects of fragmentation on infectious disease dynamics. *Science.*, 344, 1289–93.
- Laine, A.-L. (2011). Context-dependent effects of induced resistance under co-infection in a plant-pathogen interaction. *Evol. Appl.*, 4, 696–707.
- Ojanen, S.P., Nieminen, M., Meyke, E., Pöyry, J. & Hanski, I. (2013). Long-term metapopulation study of the Glanville fritillary butterfly (*Melitaea cinxia*): Survey methods, data management, and long-term population trends. *Ecol. Evol.*, 3, 3713–3737.
- Ovaskainen, O. & Soininen, J. (2011). Making more out of sparse data: hierarchical modeling of species communities. *Ecology*, 92, 289–295.
- Ovaskainen, O., Tikhonov, G., Norberg, A., Blanchet, F.G., Duan, L., Dunson, D.B., *et al.* (2017). How to make more out of community data? A conceptual framework and its implementation as models and software. *Ecol. Lett.*, 2, 561–576.
- Telfer, S., Lambin, X., Britles, R., Beldomenico, P., S., B., Paterson, S., *et al.* (2010). Species Interactions in a Parasite Community Drive Infection Risk in a Wildlife Population. *Science*, 330, 243–247.
- Tjur, T. (2009). Coefficients of Determination in Logistic Regression Models—A New Proposal: The Coefficient of Discrimination. *Am. Stat.*, 63, 366–372.
- Veech, J.A. (2013). A probabilistic model for analysing species co-occurrence. *Glob. Ecol. Biogeogr.*, 22, 252–260.
- Warton, D.I., Blanchet, F.G., O'Hara, R.B., Ovaskainen, O., Taskinen, S., Walker, S.C., *et al.* (2015). So Many Variables: Joint Modeling in Community Ecology. *Trends Ecol. Evol.*, 30, 766–779.